# YOUR AUTOREGRESSIVE GENERATIVE MODEL CAN BE BETTER IF YOU TREAT IT AS AN ENERGY-BASED ONE

## ABSTRACT

Autoregressive generative models are commonly used, especially for those tasks involving sequential data. They have, however, been plagued by a slew of inherent flaws due to the intrinsic characteristics of chain-style conditional modeling (e.g., exposure bias or lack of long-range coherence), severely limiting their ability to model distributions properly. In this paper, we propose a unique method for training the autoregressive generative model that takes advantage of a well-designed energy-based learning objective. We show that our method is capable of alleviating the exposure bias problem and increase temporal coherence by imposing a constraint which fits joint distributions at each time step. Besides, unlike former energy-based models, we estimate energy scores based on the underlying autoregressive network itself, which does not require any extra network. Finally, thanks to importance sampling, we can train the entire model efficiently without requiring an MCMC process. Extensive empirical results, covering benchmarks like language modeling, neural machine translation, and image generation, demonstrate the effectiveness of the proposed approach.

## 1 INTRODUCTION

By factorizing the joint distribution into the product of a series of conditional distributions, autoregressive generative models (abbr. ARGMs) (Vaswani et al., 2017; Dai et al., 2019; van den Oord et al., 2016a;b; Salimans et al., 2017; Chen et al., 2018) simplify the difficult challenge of modeling high-dimensional joint distributions. They can be trained efficiently via maximum likelihood and generate samples of exceptional quality, making this technique popular for modeling distributions, especially for sequential data. Nonetheless, despite their potency and flexibility, ARGMs still have inherent weaknesses due to the intrinsic characteristics of chain-style conditional modeling. For example, ARGMs usually suffer from a discrepancy of the input context distributions between the training and inference stages, which causes consequent error propagation (*i.e.*, Exposure Bias (Ranzato et al., 2016; Bengio et al., 2015)). Besides, due to the nature of greedy selection of beam search approximations, the decoded results from ARGMs may also lack in long-range coherence. We consider one approach by which ARGMs could be adapted to reduce these concerns.

Earlier work, both heuristic and theoretical, has already been proposed with those goals. For instance, the exposure bias problem of ARGMs can be alleviated to some extent with scheduled sampling (Bengio et al., 2015; Mihaylova & Martins, 2019), by mixing input contexts from both real data and autoregressive generation, during the training stage. However, this scheme suffers from an over-correcting problem (Zhang et al., 2019). In addition, at the inference stage, beam search makes it possible to choose more diverse candidates, improving the quality of generated sequences. Nevertheless, this results in only marginal improvements in temporal coherence, since ARGMs can only leverage previous decoded contexts without consideration of the whole sequence information. Moreover, setting aside the difficulty in training them, energy-based models (EBMs) have demonstrated their effectiveness in modeling high-dimensional distributions in a variety of machine learning applications (Zhao et al., 2017; Arbel et al., 2021; Gao et al., 2021), without requiring the transformation of the target distribution into a product of conditional distributions. As a result, several studies (Deng et al., 2020; Bakhtin et al., 2021; Durkan & Nash, 2019) attempt to combine EBMs with ARGMs, expecting to benefit from the strengths of both approaches. However, though some positive results were obtained, the existing works preferred a two-stage optimization, which first obtains a well-trained ARGM and then trains an additional EBM based on it. Such an optimiza-

tion strategy does not enable ARGMs to benefit from the properties of EBM in modeling the joint distribution in a temporally more coherent way.

In this paper, we present a novel design for seamlessly integrating **E**nergy-based models into **A**uto**R**egressive **M**odels (E-ARM). Our training is based on an energy-based learning objective, which forces ARGMs training to fit the joint distribution along with the conditional one at each time step. Thanks to our well-designed energy function, the two involved models can share a single base network without additional parameters, that is, the base network not only serves as a generator that provides fake data to facilitate the training of EBMs like previous works (Che et al., 2020; Xiao et al., 2021; Durkan & Nash, 2019; Deng et al., 2020), but also plays the role of modeling the energy surface. This property makes it easy to plug E-ARM into the training of any autoregressive generative models.

Intuitively, the exposure bias in ARGMs is caused by the fact that the model is trained on real data rather than data generated by the model. On the other hand, in the EBM's optimization process for modeling joint densities, the negative phase of wake-sleep algorithms (Hinton, 2002; Kim & Bengio, 2016) requires sampling data from the EBM itself. Along with the fact that our method combines the EBM and the ARGM seamlessly as a whole, E-ARM can reduce the discrepancy between input data of the training and inference stage, which mitigates the exposure bias problem of the ARGM. On top of it, unlike ARGMs, which factor the joint distribution into a product of conditional distributions, EBMs are able to model the joint distribution directly and score each input at the sequence level instead of at the token level, which makes them capable of modeling long-range coherence. Additionally, in order to optimize the proposed energy-based learning objective efficiently via gradient-based wake-sleep algorithms (Kim & Bengio, 2016), we present a way to estimate the negative phase gradient (which is a necessary component in the gradient-based wake-sleep algorithms) through those samples generated with the autoregressive view instead of the EBM view, which would require an expensive Markov Chain Monte Carlo (MCMC) process. This allows us to sidestep extremely time-consuming MCMCs, thus accelerating training.

In summary, the following contributions are made with this paper: i) We introduce a novel scheme, E-ARM, to integrate the EBM view into autoregressive generative models seamlessly; ii) we attempt to reduce the intrinsic problems of autoregressive models, such as exposure bias and weak temporal coherence, by optimizing an energy-based learning objective, which uses samples autoregressively generated; iii) We demonstrate how to efficiently optimize our model constructed from a single network, using wake-sleep algorithms without MCMC; iv) In a number of applications, such as language modeling, neural machine translation, and image generation, our model can achieve better results in comparison with relevant baselines.

## 2 BACKGROUND

### 2.1 ENERGY-BASED MODELS

Energy-based models (LeCun et al., 2006) can express any probability $p(\mathbf{x})$ for $\mathbf{x} \in \mathbb{R}^K$ as

$$p_\theta(\mathbf{x}) = \frac{\exp(-\mathbf{E}_\theta(\mathbf{x}))}{\mathbf{Z}_\theta}, \tag{1}$$

where $E_\theta : \mathbb{R}^D \to \mathbb{R}$ denotes an energy function which aims to map a $D$-dimensional datapoint to a scalar, and $\mathbf{Z}(\theta) = \sum_{\mathbf{x}} \exp(-\mathbf{E}_\theta(\mathbf{x}))$ denotes the normalizing constant, also known as the partition function. Any function can be used as an energy function to represent an EBM as long as it can generate a single scalar given some input $\mathbf{x}$ and the normalizing constant is finite[1]. Wake-sleep algorithms are commonly used to optimize EBMs (Hinton, 2002; Kim & Bengio, 2016; Grathwohl et al., 2020) via gradient-based approximate maximum likelihood. Specifically, the gradient of the log-likelihood, which needs to be maximized, with respect to $\theta$ can be expressed as

$$\mathbb{E}_{p_d(\mathbf{x})}\left[\frac{\partial}{\partial \theta} \log p_\theta(\mathbf{x})\right] = \mathbb{E}_{p_\theta(\mathbf{x})}\left[\frac{\partial}{\partial \theta}\mathbf{E}_\theta(\mathbf{x})\right] - \mathbb{E}_{p_d(\mathbf{x})}\left[\frac{\partial}{\partial \theta}\mathbf{E}_\theta(\mathbf{x})\right]. \tag{2}$$

---

[1]Without constraining the parametrization of $\mathbf{E}_\theta$, this can be achieved by bounding the region of space in which $x$ takes its allowed values.

The first term in the right hand side of Eq. 2 is the negative phase term while the second term is called the positive phase term. MCMC methods have been used (Hinton, 2002; Welling & Teh, 2011a) to approximately sample from $p_\theta(\mathbf{x})$, for estimating the negative phase term.

## 2.2 MODELING DISTRIBUTIONS AUTOREGRESSIVELY

Autoregressive generative models (ARGM)[2] can decompose any joint distribution $p(\mathbf{x})$ into a product of conditional distributions using the product rule of probability by ordering those random variables within the joint distribution and characterizing each random variable given all variables preceding it in that order. Formally, we use $\mathbf{x}_{<k}$ to denote the vector variable covering all random variables before the time step $k$ and use $x_k$ denote the random variable at time step $k$. Then we have

$$p(\mathbf{x}) = \prod_{k=1}^{K} p(x_k|\mathbf{x}_{<k}). \tag{3}$$

In general, modeling distributions autoregressively has achieved remarkable accomplishments in numerous areas (Vaswani et al., 2017; Radford et al., 2019; van den Oord et al., 2016c;b; Salimans et al., 2017) thanks to its ability to avoid the challenging target of modeling joint high-dimensional distributions directly. We primarily focus on autoregressive language models in this paper, but we also conduct experiments on image generation to validate the generality of our method.

## 2.3 EXPOSURE BIAS AND INCOHERENCE PROBLEMS IN AUTOREGRESSIVE MODELS

In the discussion about the defects of sequential autoregressive generative models, the exposure bias problem (Bengio et al., 2015; Ranzato et al., 2016) is an important issue, which greatly affects the model's deployment performance. During the training stage, the autoregressive model is always conditioned on ground truth token sequences. In generation stage, however, the model has to rely on its own previously generated tokens to predict the next token, when the model is deployed. If an incorrect token is selected, this error can be amplified in following steps because the next prediction will be made using an unusual input (one unlike those in the training set). Besides, out of the consideration of efficiency, autoregressive decoding usually selects the most probable token at each time step, given the ones previously selected. Such a scheme assumes the largest joint probability of the whole sequence can be achieved by separately choosing the most probable next token (given its previous context) over all time steps, which is only the local optimum. Correspondingly, the chosen sequence can not always be the model's optimum result.

## 3 INTEGRATE EBMS INTO AUTOREGRESSIVE MODELS SEAMLESSLY

For a long time, as a result of compromises for improving training stability and efficiency (*e.g.*, modeling a joint distribution by decomposing it and using a teacher-forcing training strategy), conventional autoregressive generative models have suffered from flaws such as the exposure bias and the lack of long-range coherence. To tackle these issues, we attempt to seamlessly integrate **E**nergy-based models into **A**uto**R**egressive **M**odels (E-ARM), which can be regarded as a variant of autoregressive generative models blending with an energy-based learning objective. Given a joint sequential distribution, E-ARM also addresses it autoregressively, that is, tackling tokens step by step under a specific order. However, what differs from conventional ARGMs is that we attempt to model both the conditional and the joint distributions simultaneously at each time step. In this way, E-ARM can model distributions conveniently in an autoregressive manner while avoiding those potential problems brought by ARGMs.

Formally, given a sequence of random variables $(x_1, x_2, \ldots, x_K)$ with length $K$, we introduce a parametric autoregressive model $q_\theta(\mathbf{x}_{<k}) = \prod_{l=1}^{k-1} q_\theta(x_l|\mathbf{x}_{<l})$ ($k$ denotes the time step) with parameters $\theta$. Particularly, we define $\tilde{q}_\theta(\mathbf{x}_{<k}) = \prod_{l=m}^{k-1} q_\theta(x_l|\mathbf{x}_{<l}) \prod_{n=1}^{m-1} q(x_n|\mathbf{x}_{<n})$, which means only those conditional distributions $q_\theta(x_l|\mathbf{x}_{<l})$ of the most recent $k - m$ time steps are involved in the current update of parameters $\theta$ while those distant conditional distributions $q(x_n|\mathbf{x}_{<n})$ are

---

[2]In this paper, the term "autoregressive model" is sometimes used to denote the autoregressive generative model for convenience.

treated as fixed (The rationale behind such a design will be elaborated in Sec.4). Then, we define $p_\theta(x_k, \mathbf{x}_{<k})$ as a product of the autoregressive model and an EBM as follows,

$$p_\theta(x_k, \mathbf{x}_{<k}) = \tilde{q}_\theta(\mathbf{x}_{<k}) \cdot \frac{e^{-\phi_\theta(x_k, \mathbf{x}_{<k})}}{\mathbf{Z}_\theta}, \tag{4}$$

where the energy function $\phi_\theta(x_k, \mathbf{x}_{<k})$ is defined as the $x_k$'s negative corresponding component of the base network's output logit with the input prefix context $\mathbf{x}_{<k} = (x_1, x_2, \ldots, x_{k-1})$ (*e.g.*, given a sequence "This is Friday." and assuming the corresponding index of the token "Friday" in the vocabulary is $i$, then the value of $-\phi_\theta$("Friday", "This is") is the $i$-th component of the output logit, which is the straight input tensor of the final softmax layer), and the normalization term $\mathbf{Z}_\theta = \mathbb{E}_{\mathbf{x}'_{<k} \sim \tilde{q}_\theta(\mathbf{x}_{<k})}[\sum_{x_k} e^{-\phi_\theta(x_k, \mathbf{x}'_{<k})}]$.

Our primary goal is to make the distribution $q_\theta(x_k|\mathbf{x}_{<k})$ to approach the real conditional $p_d(x_k|\mathbf{x}_{<k})$ whilst maintaining $p_\theta(x_k, \mathbf{x}_{<k})$ as close to the real joint $p_d(x_k, \mathbf{x}_{<k})$ as possible at each time step, which can be achieved by minimizing the Kullback-Leibler (KL) divergence between the distributions,

$$\theta^* = \arg\min_\theta \sum_{k=1}^K \left[ \mathbf{D}_{KL}\Big(p_d(x_k|\mathbf{x}_{<k})||q_\theta(x_k|\mathbf{x}_{<k})\Big) + \lambda\mathbf{D}_{KL}\Big(p_d(x_k, \mathbf{x}_{<k})||p_\theta(x_k, \mathbf{x}_{<k})\Big) \right], \tag{5}$$

where $\lambda$ adjusts the ratio between the two objectives. In Eq. 5, the first objective at each time step $k$ can be easily optimized by cross entropy while the second objective is optimized in the sense of EBMs by wake-sleep algorithms (Hinton et al., 1995; Kim & Bengio, 2016), which minimizes the objective by descending the following gradient of $\theta$ according to Eq. 2[3]

$$\underbrace{\mathbb{E}_{x_k, \mathbf{x}_{<k} \sim p_d(x_k, \mathbf{x}_{<k})}\left[\frac{\partial}{\partial\theta}\mathbf{E}_\theta(x_k, \mathbf{x}_{<k})\right]}_{\textbf{Positive Phase}} - \underbrace{\mathbb{E}_{x_k, \mathbf{x}_{<k} \sim p_\theta(x_k, \mathbf{x}_{<k})}\left[\frac{\partial}{\partial\theta}\mathbf{E}_\theta(x_k, \mathbf{x}_{<k})\right]}_{\textbf{Negative Phase}}, \tag{6}$$

where we have $\mathbf{E}_\theta(x_k, \mathbf{x}_{<k}) = \phi_\theta(x_k, \mathbf{x}_{<k}) - \log \tilde{q}_\theta(\mathbf{x}_{<k})$. Optimization via Eq. 2 or 6 involves sampling data from the model and can thus lead to the discovery of non-data-like samples, whose likelihood is then explicitly reduced by the energy function. E-ARM is therefore not plagued by the exposure bias problem. Besides, because we model the joint distribution throughout the training process, E-ARM can assess the entire sequence as a whole and generate more coherent data using energy sampling (Deng et al., 2020).

## 4 OPTIMIZATION

In this section, we present how to efficiently optimize E-ARM. To begin with, we optimize the first objective in Eq. 5 as with conventional autoregressive models by reducing the per time-step cross-entropy loss. As for the second objective, we resort to descend the estimated gradient as shown in Eq. 6. Thanks to the importance sampling technique and our well-defined energy function, we now show that an improved version of Eq. 6 has a simple and symmetric form that can be easily estimated whilst not requiring an expensive MCMC.

Specifically, by replacing $\mathbf{E}_\theta(x_k, \mathbf{x}_{<k})$ with the specific form $\phi_\theta(x_k, \mathbf{x}_{<k}) - \log \tilde{q}_\theta(\mathbf{x}_{<k})$, the gradient w.r.t. $\theta$ in the positive phase of Eq. 6 can be written into

$$-\mathbb{E}_{\mathbf{x}_{<k} \sim p_d}[\frac{\partial}{\partial\theta}\log\tilde{q}_\theta(\mathbf{x}_{<k})] + \mathbb{E}_{x_k, \mathbf{x}_{<k} \sim p_d}[\frac{\partial}{\partial\theta}\phi_\theta(x_k, \mathbf{x}_{<k})]. \tag{7}$$

Similarly, we can get the negative phase gradient as

$$-\mathbb{E}_{\mathbf{x}_{<k} \sim p_\theta}[\frac{\partial}{\partial\theta}\log\tilde{q}_\theta(\mathbf{x}_{<k})] + \mathbb{E}_{x_k, \mathbf{x}_{<k} \sim p_\theta}[\frac{\partial}{\partial\theta}\phi_\theta(x_k, \mathbf{x}_{<k})]. \tag{8}$$

The first term $-\mathbb{E}_{\mathbf{x}_{<k} \sim p_d}[\frac{\partial}{\partial\theta}\log\tilde{q}_\theta(\mathbf{x}_{<k})]$ in Eq. 7 is equivalent to the log-likelihood gradient of $\tilde{q}_\theta(\mathbf{x}_{<k})$, which means improvements in this direction will be automatically taken care of as a result of steps arising from the gradient of the first KL-divergence in Eq. 5, albeit at the expense of

---

[3]here, we take a minimization version of the Eq. 2. Thus the sign before each phase is converse.

changing the weight given to the second vs. the first KL, $\lambda$. Besides, because the estimation of the expectation operator over the data distribution $p_d$ is easy, and the score $\phi_\theta(x_k, \mathbf{x}_{<k})$ can be acquired simply accessing the output logit of ARGM (see the definition of $\phi_\theta$ in Sec. 3), the second term can likewise be readily estimated and optimized. As a result, the positive phase optimization is both feasible and efficient.

The negative phase gradient estimation, on the other hand, is more involved. In Eq. 8, sampling data from $p_\theta$ is required for estimating the expectation $\mathbb{E}_{p_\theta}$, whereas $p_\theta$ is a parametric joint probability involving an energy-based unnormalized probability estimator that may require time-consuming MCMC methods to generate data. However, thanks to importance sampling, we can substitute the troublesome computation of the expectation over the distribution $p_\theta$ with the expectation over the distribution $q_\theta$, which can generate samples autoregressively without MCMC. Formally, the negative phase gradient $\mathbb{E}_{x_k, \mathbf{x}_{<k} \sim p_\theta}[\frac{\partial}{\partial\theta} \mathbf{E}_\theta(x_k, \mathbf{x}_{<k})]$ is equivalent to the following formulation (See the detailed derivation in Appendix A),

$$-\mathbb{E}_{\mathbf{x}_{<k} \sim \tilde{q}_\theta(\mathbf{x}_{<k})}[\mathbf{w}(\mathbf{x}_{<k})\frac{\partial}{\partial\theta} \log \tilde{q}_\theta(\mathbf{x}_{<k})] + \mathbb{E}_{x_k, \mathbf{x}_{<k} \sim \tilde{q}_\theta(x_k, \mathbf{x}_{<k})}[\mathbf{w}(\mathbf{x}_{<k})\frac{\partial}{\partial\theta} \phi_\theta(x_k, \mathbf{x}_{<k})], \quad (9)$$

$$\text{where} \quad \mathbf{w}(\mathbf{x}_{<k}) = \frac{\sum_{x_k} e^{-\phi(x_k, \mathbf{x}_{<k})}}{\mathbb{E}_{\mathbf{x}'_{<k} \sim \tilde{q}_\theta(\mathbf{x}_{<k})}[\sum_{x_k} e^{-\phi_\theta(x_k, \mathbf{x}'_{<k})}]}. \quad (10)$$

According to Eq. 9, all the estimated expectations only need sampling from the autoregressive model rather than the joint distribution, and the reweighing weight $\mathbf{w}$ in Eq. 10 also does not involve expectation computation over distribution $p_\theta$. Generally, producing data from an autoregressive model is a very simple ancerstral sampling process, as compared with sampling straight from an EBM, which needs MCMC approaches (Durkan & Nash, 2019). On account of that, the optimization process can be much more efficient.

Besides, the term $\mathbb{E}_{\mathbf{x}_{<k} \sim \tilde{q}_\theta(\mathbf{x}_{<k})}[\mathbf{w}(\mathbf{x}_{<k})\frac{\partial}{\partial\theta} \log \tilde{q}_\theta(\mathbf{x}_{<k})]$ in Eq. 9 is equivalent to a re-weighted version of the gradient of $q_\theta$'s information entropy with respect to $\theta$. This term can be optimized similarly to the teacher-forcing training of autoregressive model with the "teacher" sequence generated autoregressively by the model itself. Actually, the scheduled sampling methods (Bengio et al., 2015; Ranzato et al., 2016; Mihaylova & Martins, 2019) are similar to this term but without the re-weighting factor. Furthermore, it is worth noting that for a sequence with total length $K$, since we add a constraint to fit the joint distribution $p_\theta$ at each time step $k$, Eq. 9 actually has $K$ counterparts with different time steps. If we use the $q_\theta(\mathbf{x}_{<k})$ directly instead of $\tilde{q}_\theta(\mathbf{x}_{<k})$ in the Eq. 4 to define $p_\theta(x_k, \mathbf{x}_{<k})$, due to the fact that the distribution $q_\theta(\mathbf{x}_{<k})$ modeled by an autoregressive model can be naturally broken up into pieces, simply summing up these $K$ gradients results in the term

$$\sum_{k=1}^{K}\mathbb{E}_{q_\theta(\mathbf{x}_{<k})}[\mathbf{w}(\mathbf{x}_{<k})\frac{\partial}{\partial\theta} \log q_\theta(\mathbf{x}_{<k})] = \sum_{l=1}^{K} \sum_{k=1}^{K+1-l} \mathbb{E}_{q_\theta(\mathbf{x}_{<k})}[\mathbf{w}(\mathbf{x}_{<k})\frac{\partial}{\partial\theta} \log q_\theta(x_l|\mathbf{x}_{<l})], \quad (11)$$

where $l$ indicates the specific index of the current token in the entire sequence. As a result, earlier time steps (smaller $l$) will get stronger training signals (larger $K + 1 - l$, indicating more gradient terms), giving rise to imbalanced training for different time steps. To solve this, we introduce $\tilde{q}_\theta(\mathbf{x}_{<k})$ as $\prod_{l=m}^{k-1} q_\theta(x_l|\mathbf{x}_{<l}) \prod_{n=1}^{m-1} q(x_n|\mathbf{x}_{<n})$ to define $p_\theta(x_k, \mathbf{x}_{<k})$ shown in Sec. 3, allowing gradients only back propagate through conditional distributions w.r.t. a few recent tokens[4]. This explains our proposal of using $\tilde{q}_\theta(\mathbf{x}_{<k})$ to define $p_\theta(x_k, \mathbf{x}_{<k})$.

Ultimately, combining Eq. 7 and Eq. 9 , at each time step $k$, we can optimize $p_\theta(x_k, \mathbf{x}_{<k})$ via descending the estimated gradient of $\theta$ as follows,

$$\underbrace{\left( \begin{array}{c} - \quad \mathbb{E}_{\mathbf{x}_{<k} \sim p_d}[\frac{\partial}{\partial\theta} \log \tilde{q}_\theta(\mathbf{x}_{<k})] \\ + \mathbb{E}_{x_k, \mathbf{x}_{<k} \sim p_d}[\frac{\partial}{\partial\theta} \phi_\theta(x_k, \mathbf{x}_{<k})] \end{array} \right)}_{\textbf{Positive Phase}} - \underbrace{\left( \begin{array}{c} - \quad \mathbb{E}_{\mathbf{x}_{<k} \sim \tilde{q}_\theta(\mathbf{x}_{<k})}[\mathbf{w}(\mathbf{x}_{<k})\frac{\partial}{\partial\theta} \log \tilde{q}_\theta(\mathbf{x}_{<k})] \\ + \mathbb{E}_{x_k, \mathbf{x}_{<k} \sim \tilde{q}_\theta(x_k, \mathbf{x}_{<k})}[\mathbf{w}(\mathbf{x}_{<k})\frac{\partial}{\partial\theta} \phi_\theta(x_k, \mathbf{x}_{<k})] \end{array} \right)}_{\textbf{Negative Phase}}.$$

$$(12)$$

---

[4]In practice, we find that using recent 2 tokens worked best.

From Eq. 12, we can see that the only difference between two phases is that in the negative phase, the expectation over $\tilde{q}_\theta$ have a reweighing weight $\mathbf{w}$ for each sample. The reweighing weight $\mathbf{w}$ in Eq. 10 and Eq. 12 can be further refined (see the derivation in Appendix B) and we can observe that

$$\mathbf{w}(\mathbf{x}_{<k}) = \frac{\mu(\mathbf{x}_{<k})}{\mathbb{E}_{\mathbf{x}'_{<k}}\mu(\mathbf{x}_{<k})}, \tag{13}$$

where $\mu(\mathbf{x}_{<k}) = \frac{p_\theta(\mathbf{x}_{<k})}{\tilde{q}_\theta(\mathbf{x}_{<k})}$ indicates the possibility of which distribution the prefix context $\mathbf{x}_{<k}$ is most likely to come from, the distribution $p_\theta$ or the distribution $\tilde{q}_\theta$. Correspondingly, $\mathbf{w}(\mathbf{x}_{<k})$ reflects the context $\mathbf{x}_{<k}$'s relative magnitude of $\mu(\mathbf{x}_{<k})$ compared with the average among all potential contexts—the larger the value of $\mathbf{w}(\mathbf{x}_{<k})$, the more likely the context $\mathbf{x}_{<k}$ in the data space coming from $p_\theta$, which is modeled by the product of autoregressive models and EBMs. During training, those input sequences with contexts more likely under $p_\theta$ than $q_\theta$ will be assigned larger weights $\mathbf{w}$ while others will be assigned smaller weights $\mathbf{w}$.

In general, E-ARM ought to be viewed as a new learning pattern for autoregressive models that ensures our base autoregressive network stays close to the real distribution $p_d$. We found that training from scratch with the energy-based learning objective of in Eq.12 alone did not work well. The reason is that at the initial stage of the training process, what we have is just a randomly initialized autoregressive network which outputs sequences with random values given any context. This indicates disjoint supports between the real sequence's distribution $p_d$ and distribution $p_\theta$ modeled by ARGMs. If we only use the energy-based learning objective of Eq. 12, the whole gradient $\mathbb{E}_{p_d(\mathbf{x})}[\frac{\partial}{\partial\theta}\log p_\theta(\mathbf{x})]$ in Eq.2 would be 0 due to disjoint supports between $p_d$ and $p_\theta$. As a result, in order to make the optimization more feasible, we must maintain the cross-entropy loss low throughout training and pre-train as a pure ARGM for a few epochs before introducing the E-ARM objective. Actually, the starting epoch of E-ARM is a hyper-parameter, and we discuss it in the Sec. 5.2.

Following the excellent work of Deng et al. (2020); Bakhtin et al. (2021), we also adopt Top-K energy re-sampling in the inference stage, which means that in the generative process, we first gather multiple candidate sequences generated autoregressively, and then re-sample from them based on their energy scores estimated by the network's logit at the last time step where the entire sequence has been processed. Since we employ the EBM to model the joint distribution at each time step, such a re-sampling strategy can mitigate the undesirable impact of the greedy selection of one token at a time, and we found this variation to increase the coherence of generated samples.

## 5 EXPERIMENTS

To empirically corroborate the effectiveness of E-ARM and show its broad applicability, we conduct extensive experiments covering three machine learning applications, which are neural machine translation (NMT), language modeling, and image generation. In this section, we will introduce the three corresponding experimental setups, followed by an analysis of the obtained results. We will release the source code once upon acceptance.

### 5.1 APPLICATION TO NEURAL MACHINE TRANSLATION

E-ARM is first evaluated in the context of neural machine translation (NMT), which is a conditional generation task and is important in the natural language processing (NLP) field. We first analyze E-ARM on the IWSLT14 dataset, which includes six different language pairs ({German, Spanish, Italian} → English and English → {German, Spanish, Italian}). In addition, we test E-ARM on the WMT16 (English → German) benchmark to make sure we evaluating E-ARM on a larger dataset. Hereafter we abbreviate English, German, Spanish, Italian as "En", "De", "Es", "It". The weight $\lambda$ in Eq. 5 is set as 0.05 for all translation tasks. We use one size of transformer ("Base-IWSLT") for the IWSLT14 benchmark and two sizes of transformer ("Base-WMT", "Large-WMT") for the WMT16 benchmark [5]. Scheduled Sampling is carried out following Mihaylova & Martins (2019).

The results of IWSLT14 tasks are shown in Table 1. We test not only the pure performance of E-ARM but also the compatibility with other techniques. Specifically, we can observe that (1) without any particular engineering, E-ARM outperforms the base autoregressive translation model

---

[5]The implementation is developed on Fairseq (Ott et al., 2019).

| Model | Label Smoothing | Scheduled Sampling | Beam Searching | BLEU ↑ | | | | | | Avg. |
|---|---|---|---|---|---|---|---|---|---|---|
| | | | | DE→EN | EN→DE | EN→IT | IT→EN | ES→EN | EN→ES | |
| **Base** | - | - | - | 32.44±0.06 | 26.64±0.10 | 27.92±0.03 | 30.48±0.08 | 38.61±0.11 | 35.42±0.09 | 31.92 |
| | - | - | 5 B | 33.62±0.07 | 27.41±0.08 | 28.72±0.04 | 31.39±0.05 | 39.55±0.12 | 36.38±0.07 | 32.85 |
| | ✔ | - | - | 33.68±0.03 | 27.62±0.04 | 28.81±0.07 | 31.42±0.07 | 39.85±0.13 | 36.71±0.09 | 33.02 |
| | ✔ | - | 5 B | 34.61±0.08 | 28.46±0.06 | 29.72±0.10 | 32.29±0.03 | 40.64±0.07 | 37.48±0.05 | 33.87 |
| | ✔ | ✔ | - | 34.23±0.06 | 27.96±0.03 | 29.26±0.11 | 31.93±0.08 | 40.16±0.03 | 37.21±0.04 | 33.46 |
| | ✔ | ✔ | 5 B | 35.10±0.04 | 28.73±0.04 | 29.97±0.07 | 32.64±0.12 | 40.91±0.06 | 37.93±0.10 | 34.21 |
| **E-ARM** | - | - | - | 32.99±0.13 | 27.15±0.03 | 28.33±0.12 | 31.13±0.04 | 39.56±0.01 | 36.07±0.02 | 32.54 |
| | - | - | 5 B | 34.06±0.06 | 27.97±0.08 | 29.26±0.09 | 31.90±0.13 | 40.30±0.03 | 36.92±0.09 | 33.40 |
| | ✔ | - | - | 33.97±0.08 | 28.03±0.04 | 29.13±0.02 | 31.84±0.11 | 40.32±0.03 | 36.96±0.07 | 33.38 |
| | ✔ | - | 5 B | 34.93±0.05 | 28.91±0.12 | 30.04±0.11 | 32.56±0.04 | 41.01±0.06 | 37.73±0.12 | 34.20 |
| | ✔ | ✔ | - | 34.58±0.09 | 28.38±0.12 | 29.56±0.10 | 32.11±0.03 | 40.93±0.03 | 37.56±0.07 | 33.85 |
| | ✔ | ✔ | 5 B | **35.36**±0.05 | **29.11**±0.04 | **30.25**±0.09 | **32.82**±0.11 | **41.58**±0.07 | **38.19**±0.03 | **34.55** |

Table 1: Comparison of BLEU scores between our approach E-ARM and the base ARGM trained just with cross-entropy loss on six translation pairs of IWSLT14 datasets. We use "-" to denote that the training trick is not used while "✔" indicates we use it. "**5 B**" represents we use beam searching with 5 beams.

trained with cross-entropy singly by 0.62 (31.92 → 32.54) BLEU points in average, especially on three translation pairs—38.61 → 39.56 on Spanish-to-English, 30.48 → 31.13 on Italian-to-English, 35.42 → 36.07 on English-to-Spanish. (2) E-ARM is compatible with other techniques like scheduled sampling, which can help alleviate the exposure bias problem to some extent. They are not mutually exclusive and can work together to further improve the performance of the base ARGM. (3) However, since scheduled sampling can reduce exposure bias and beam search can somewhat alleviate the flaws caused by greedy selection at each time step, the performance gain of E-ARM when all these tactics are combined is only 0.34 (34.21 → 34.55), which is lower than the 0.62 (31.92 → 32.54) obtained when the model is purely trained without these other techniques.

| Model | L.S. | S.S. | w/E-ARM | BLEU ↑ |
|---|---|---|---|---|
| **Base-WMT** | - | - | - | 27.56 |
| | ✔ | - | - | 28.04 |
| | ✔ | ✔ | - | 28.36 |
| | ✔ | ✔ | ✔ | **28.62** |
| **Large-WMT** | - | - | - | 28.70 |
| | ✔ | - | - | 29.05 |
| | ✔ | ✔ | - | 29.23 |
| | ✔ | ✔ | ✔ | **29.44** |

Table 2: Translation performance of proposed E-ARM on WMT16 English→German, evaluated with BLEU. We uniformly use 5 beams when applying beam search. "**L.S.**" denotes Label Smoothing and "**S.S.**" denotes Scheduled Sampling.

Additionally, Table 2 shows the performance of E-ARM on the WMT16 English → German task. For two different model sizes, enabling label smoothing (L.S.) improves model performance by 0.52 and 0.35, respectively. The performance of the base transformer model further increases to 28.36 BLEU points when scheduled sampling (S.S.) is used, while the larger model improves to 29.23 points. E-ARM paired with label smoothing and scheduled sampling yields the highest scores of 28.62 and 29.44, respectively. Overall, our training strategy outperforms ARGM's vanilla teacher-forcing training and can have uniformly favorable impacts across different models and dataset sizes.

## 5.2 APPLICATION TO LANGUAGE MODELING

To further demonstrate E-ARM's consistency in reducing flaws of autoregressive generative models, we also conduct language modeling experiments. The WikiText-103 dataset (Merity et al., 2017), which is the largest word-level language modeling benchmark with long-term dependency, was chosen as the testbed. It comprises 103 million

| Model | #Params | PPL ↓ |
|---|---|---|
| **Tr-Base** | 156M | 30.56 |
| **Tr-Base (w/E-ARM)** | 156M | **29.89** |
| **Standard Tr-XL** | 151M | 24.20 |
| **Standard Tr-XL (w/E-ARM)** | 151M | **23.81** |

Table 3: Language modeling performance of different models on WikiText103. Evaluation is conducted using perplexity (PPL).

training tokens from 28 thousand articles, with an average length of 3.6 thousand tokens per article, which allows model to evaluate the ability of modeling long-term dependency. Two network structures are mainly tested, which are Transformer-Base (Vaswani et al., 2017) and Transformer-XL (Dai et al., 2019) (Tr-Base and Tr-XL for short respectively hereafter).

The final results are reported in Table 3. We can see from the results that E-ARM outperforms baselines with clear margins for different types of models. Specifically, the Transformer-Base improves performance by 0.67 PPL points (from 30.56 to 29.89), while the Transformer-XL improves model by 0.20 PPL points (from 24.20 to 23.81). Our strategy does not change the structure of the base network nor introduces any additional module or learnable parameters, therefore we can conclude that the performance boost is solely from the introduced energy-based learning objective.

|  |  | **Start Epoch** | | |
|---|---|---|---|---|
|  |  | **5** | **15** | **25** |
|  | **0.00** | 30.56 | 30.56 | 30.56 |
|  | **0.01** | 30.48 | 30.12 | 30.22 |
| $\lambda$ | **0.05** | 30.43 | **29.89** | 30.16 |
|  | **0.1** | 30.60 | 30.03 | 30.14 |
|  | **0.5** | 30.71 | 30.36 | 30.47 |

Table 4: How different $\lambda$ and the E-ARM start epoch (when we introduce the E-ARM into the training on WikiText103) affect performance evaluated by perplexity (PPL). The Tr-Base model structure is used and is train 40 epochs in total.

In addition, we study the effect of hyper-parameter settings on the performance of language modeling, which can be seen in Table 4. From this, we may deduce that starting E-ARM training at the 15-th epoch yields the best results, whereas starting earlier or later yields a performance decline. It is reasonable because, if E-ARM was introduced too early, the autoregressive model may not have been optimized well at that moment. As a result, generative quality would be terrible, and make energy-based training unstable. On the other hand, the underlying autoregressive model can be modified only marginally if E-ARM is introduced when the ARGM training is virtually complete. Besides, from the vertical perspective which presents the impact of different $\lambda$, we can observe that the best $\lambda$ in Eq. 5 is 0.05. The first line of the table indicates the baseline of training the autoregressive model with pure cross-entropy loss.

## 5.3 APPLICATION TO IMAGE GENERATION

In order to illustrate the effectiveness and generality of our method in processing different modality tasks, we further show the results of applying E-ARM to image generation in this section. We apply E-ARM to Pixel-CNN (Van Oord et al., 2016) and its variant Gated Pixel-CNN (Oord et al., 2016). Experiments are carried out on the MNIST and CIFAR-10 datasets.

| **Model** | **Test (Train) NLL ↓** | |
|---|---|---|
|  | **MNIST** | **CIFAR-10** |
| **Pixel-CNN** | 0.17 (0.13) | 3.14 (3.08) |
| **Pixel-CNN (w/E-ARM)** | **0.15 (0.12)** | **3.07 (2.98)** |
| **Gated Pixel-CNN** | 0.14 (0.11) | 3.03 (2.90) |
| **Gated Pixel-CNN (w/E-ARM)** | **0.12 (0.10)** | **2.97 (2.91)** |

Table 5: Performance of E-ARM with different base networks on MNIST and CIFAR-10 in bits/dim (lower is better), training performance in brackets.

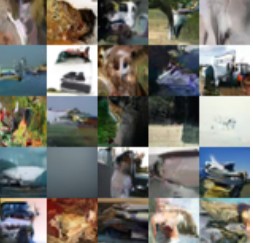

Figure 1: Samples of CIFAR-10 from Gated Pixel-CNN (w/E-ARM).

Table 5 summarizes the quantitative results measured by per-pixel negative log-likelihood (NLL), while Figure 1 depicts some of the generated samples. We can see that with the help of our E-ARM, both the Pixel-CNN and the Gated Pixel-CNN can obtain improvements in all datasets (0.17 → 0.15 and 3.14 → 3.07 for Pixel-CNN on MNIST and CIFAR10 respectively and 0.14 → 0.12 and 3.03 → 2.97 for Gated Pixel-CNN on MNIST and CIFAR10 respectively). This is further evidence in favour of the energy-based learning objective for improving autoregressive models.

## 6 RELATED WORKS

### 6.1 AUTOREGRESSIVE GENERATIVE MODELS

Modeling high-dimensional data distributions directly is usually a rather challenging task due to "the curse of dimensionality" (Bellman, 1954). One alternative method is to sequentialize the random variables and then factorize the joint probability distribution into the product of conditionals based on the sequence structure, which is exactly the core idea of autoregressive generative models (ARGMs).

ARGMs have been very successful, in particular for sequential data. For example, ARGMs have been widely used in language modeling (Vaswani et al., 2017; Dai et al., 2019; Radford et al., 2019), audio synthesis (van den Oord et al., 2016a), and even image generation (van den Oord et al., 2016c;b; Salimans et al., 2017). The advantages of ARGMs are however balanced by issues of (1) exposure bias (Ranzato et al., 2016; Bengio et al., 2015; Song et al., 2020), due to the discrepancy in input context distributions between the training and inference stages, and (2) weak long-range coherence, due to the inherent greedy selection of one token at a time without look-ahead.

### 6.2 ENERGY-BASED MODELS

In the field of generative modeling, energy-based models (EBMs) have been widely used (Zhao et al., 2017; Arbel et al., 2021; Gao et al., 2021). The primary idea behind EBMs is to decompose the dependencies between variables (*e.g.* images and labels) through different terms of an energy function, assigning low energies to proper configurations found in the dataset, while assigning high energies to incorrect or unseen ones (LeCun et al., 2006).

Due to the challenge of sampling from EBMs, training EBMs by wake-sleep algorithms (Hinton, 2002; Kim & Bengio, 2016; Grathwohl et al., 2021), which require expensive MCMC approaches, has been notoriously difficult, especially on high-dimensional data like images or texts. Stochastic Gradient Langevin Dynamics (SGLD) (Welling & Teh, 2011a) is a frequently used gradient-based MCMC approach that injects noise into parameter updates and anneals the step size during the course of training, and which has been adopted in numerous prior works (Nijkamp et al., 2019; Du & Mordatch, 2019; Grathwohl et al., 2020). However, these gradient-based MCMC methods require enormous extra computing overheads and are not applicable when the input is discrete like for text sequences (Deng et al., 2020).

As a result, a variety of recent works attempt to explore the strategy of training an EBM without MCMC. In particular, Bakhtin et al. (2021); Xu et al. (2021a); Gao et al. (2020) optimize the EBMs by using noise contrastive estimation (NCE) (Gutmann & Hyvärinen, 2010; Ma & Collins, 2018). Durkan & Nash (2019) estimate the intractable normalization component by utilizing ARGMs and importance sampling. Che et al. (2020); Wang et al. (2021) skirt the challenge of collecting data in the high-dimensional data space by producing data in the lower-dimensional feature space, which improves sampling efficiency.

## 7 CONCLUSIONS AND FUTURE WORK

In this paper, we propose a novel method dubbed E-ARM to integrate energy-based models into autoregressive generative models seamlessly, with an energy-based training objective that exploits an underlying autoregressive model. This is achieved by defining the energy function from the output logits of the base autoregressive network, to model the unnormalized joint distribution of the subsequence up to each time step. We also found ways to improve training of E-ARM using importance sampling, avoiding the requirement of MCMC for the energy-based training. Experimental results on two language tasks and one vision task demonstrate the effectiveness of E-ARM to alleviate exposure bias and incoherence problems of ARGMs. In the future, we expect to extend E-ARM on other sequential generation tasks (*e.g.* text summarization, audio generation), and incorporate the proposed methodology into other advanced autoregressive architectures.

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

## A   THE DERIVATION OF THE NEGATIVE PHASE GRADIENT

In this section, we show the detailed derivation of Eq. 9. Formally, as shown in Sec. 3, given an autoregressive model $q_\theta(\mathbf{x}_{<k}) = \prod_{l=1}^{k-1} q_\theta(x_l|\mathbf{x}_{<l})$ ($k$ denotes the time step) with parameters $\theta$, we define a product of the autoregressive model and an EBM as follows

$$p_\theta(x_k, \mathbf{x}_{<k}) = \tilde{q}_\theta(\mathbf{x}_{<k}) \cdot \frac{e^{-\phi_\theta(x_k, \mathbf{x}_{<k})}}{\mathbf{Z}_\theta}, \tag{14}$$

where $\tilde{q}_\theta(\mathbf{x}_{<k}) = \prod_{l=m}^{k-1} q_\theta(x_l|\mathbf{x}_{<l}) \prod_{n=1}^{m-1} q(x_n|\mathbf{x}_{<n})$. Under such definition, only those conditional distributions $q_\theta(x_l|\mathbf{x}_{<l})$ of the most recent $k - m$ time steps are involved in the current update of parameters $\theta$ while those distant conditional distributions $q(x_n|\mathbf{x}_{<n})$ are treated as fixed. We have explained the rationale and intuition in Sec.4. $\mathbf{Z}_\theta$ is the normalization term and equal to $\mathbb{E}_{\mathbf{x}'_{<k} \sim \tilde{q}_\theta(\mathbf{x}_{<k})}[\sum_{x_k} e^{-\phi_\theta(x_k, \mathbf{x}'_{<k})}]$. The optimization of $p_\theta(x_k, \mathbf{x}_{<k})$ includes two phases, and the gradient w.r.t $\theta$ of negative phase is

$$-\mathbb{E}_{\mathbf{x}_{<k} \sim p_\theta}[\frac{\partial}{\partial \theta} \log \tilde{q}_\theta(\mathbf{x}_{<k})] + \mathbb{E}_{x_k, \mathbf{x}_{<k} \sim p_\theta}[\frac{\partial}{\partial \theta} \phi_\theta(x_k, \mathbf{x}_{<k})]. \tag{15}$$

Next, we will show the specific derivation of these two terms in Eq. 15 so that the entire Eq. 15 can be transformed into Eq. 9.

### A.1   THE DERIVATION OF THE FIRST TERM

The first term $\mathbb{E}_{\mathbf{x}_{<k} \sim p_\theta}[\frac{\partial}{\partial \theta} \log \tilde{q}_\theta(\mathbf{x}_{<k})]$ can be processed as follows

$$\begin{aligned}
\mathbb{E}_{\mathbf{x}_{<k} \sim p_\theta}[\frac{\partial}{\partial \theta} \log \tilde{q}_\theta(\mathbf{x}_{<k})] &= \sum_{\mathbf{x}_{<k}} p_\theta(\mathbf{x}_{<k}) \frac{\partial}{\partial \theta} \log \tilde{q}_\theta(\mathbf{x}_{<k}) \\
&= \sum_{\mathbf{x}_{<k}} \sum_{x_k} p_\theta(x_k, \mathbf{x}_{<k}) \frac{\partial}{\partial \theta} \log \tilde{q}_\theta(\mathbf{x}_{<k}) \\
&= \sum_{\mathbf{x}_{<k}} \tilde{q}_\theta(\mathbf{x}_{<k}) \frac{\sum_{x_k} e^{-\phi_\theta(x_k, \mathbf{x}_{<k})}}{\mathbf{Z}_\theta} \frac{\partial}{\partial \theta} \log \tilde{q}_\theta(\mathbf{x}_{<k}) \\
&= \mathbb{E}_{\mathbf{x}_{<k} \sim \tilde{q}_\theta(\mathbf{x}_{<k})}[\mathbf{w}(\mathbf{x}_{<k}) \frac{\partial}{\partial \theta} \log \tilde{q}_\theta(\mathbf{x}_{<k})],
\end{aligned} \tag{16}$$

where we have $\mathbf{w}(\mathbf{x}_{<k}) = \frac{\sum_{x_k} e^{-\phi(x_k, \mathbf{x}_{<k})}}{\mathbb{E}_{\mathbf{x}'_{<k} \sim \tilde{q}_\theta(\mathbf{x}_{<k})}[\sum_{x_k} e^{-\phi_\theta(x_k, \mathbf{x}'_{<k})}]}$ because

$$\begin{aligned}
\mathbf{w}(\mathbf{x}_{<k}) &= \frac{\sum_{x_k} e^{-\phi(x_k, \mathbf{x}_{<k})}}{\mathbf{Z}_\theta} = \frac{\sum_{x_k} e^{-\phi(x_k, \mathbf{x}_{<k})}}{\sum_{\mathbf{x}_{<k}} \sum_{x_k} \tilde{q}_\theta(\mathbf{x}_{<k}) e^{-\phi_\theta(x_k, \mathbf{x}_{<k})}} \\
&= \frac{\sum_{x_k} e^{-\phi(x_k, \mathbf{x}_{<k})}}{\sum_{\mathbf{x}_{<k}} \tilde{q}_\theta(\mathbf{x}_{<k}) \sum_{x_k} e^{-\phi_\theta(x_k, \mathbf{x}_{<k})}} \\
&= \frac{\sum_{x_k} e^{-\phi(x_k, \mathbf{x}_{<k})}}{\mathbb{E}_{\mathbf{x}_{<k} \sim \tilde{q}_\theta(\mathbf{x}_{<k})}[\sum_{x_k} e^{-\phi_\theta(x_k, \mathbf{x}_{<k})}]}.
\end{aligned} \tag{17}$$

## A.2 THE DERIVATION OF THE SECOND TERM

Then, we tackle the second term $\mathbb{E}_{x_k,\mathbf{x}_{<k}\sim p_\theta}[\frac{\partial}{\partial\theta}\phi_\theta(x_k,\mathbf{x}_{<k})]$ as follows

$$
\begin{aligned}
\mathbb{E}_{p_\theta}\big[\frac{\partial}{\partial\theta}\phi_\theta(x_k,\mathbf{x}_{<k})\big] &= \sum_{x_k,\mathbf{x}_{<k}} p_\theta(x_k,\mathbf{x}_{<k})\frac{\partial}{\partial\theta}\phi_\theta(x_k,\mathbf{x}_{<k})\\
&= \sum_{x_k,\mathbf{x}_{<k}} p_\theta(x_k,\mathbf{x}_{<k})\frac{\tilde{q}_\theta(x_k,\mathbf{x}_{<k})}{\tilde{q}_\theta(x_k,\mathbf{x}_{<k})}\frac{\partial}{\partial\theta}\phi_\theta(x_k,\mathbf{x}_{<k})\\
&= \sum_{x_k,\mathbf{x}_{<k}} \tilde{q}_\theta(x_k,\mathbf{x}_{<k})\frac{\tilde{q}_\theta(\mathbf{x}_{<k})\cdot e^{-\phi_\theta(x_k,\mathbf{x}_{<k})}}{\mathbf{Z}_\theta\cdot\tilde{q}_\theta(x_k,\mathbf{x}_{<k})}\frac{\partial}{\partial\theta}\phi_\theta(x_k,\mathbf{x}_{<k})\\
&= \mathbb{E}_{x_k,\mathbf{x}_{<k}\sim\tilde{q}_\theta(x_k,\mathbf{x}_{<k})}\big[\frac{e^{-\phi_\theta(x_k,\mathbf{x}_{<k})}}{\tilde{q}_\theta(x_k|\mathbf{x}_{<k})}\cdot\frac{1}{\mathbf{Z}_\theta}\frac{\partial}{\partial\theta}\phi_\theta(x_k,\mathbf{x}_{<k})\big]\\
&= \sum_{\mathbf{x}_{<k}}\tilde{q}_\theta(\mathbf{x}_{<k})\sum_{x_k}\tilde{q}_\theta(x_k|\mathbf{x}_{<k})\frac{e^{-\phi_\theta(x_k,\mathbf{x}_{<k})}}{\tilde{q}_\theta(x_k|\mathbf{x}_{<k})}\cdot\frac{1}{\mathbf{Z}_\theta}\frac{\partial}{\partial\theta}\phi_\theta(x_k,\mathbf{x}_{<k})\\
&= \sum_{\mathbf{x}_{<k}}\tilde{q}_\theta(\mathbf{x}_{<k})\sum_{x_k}e^{-\phi_\theta(x_k,\mathbf{x}_{<k})}\cdot\frac{1}{\mathbf{Z}_\theta}\frac{\partial}{\partial\theta}\phi_\theta(x_k,\mathbf{x}_{<k})\\
&= \mathbb{E}_{\tilde{q}_\theta(\mathbf{x}_{<k})}\big[\sum_{x_k}\frac{e^{-\phi_\theta(x_k,\mathbf{x}_{<k})}}{\mathbf{Z}_\theta}\frac{\partial}{\partial\theta}\phi_\theta(x_k,\mathbf{x}_{<k})\big]\\
&= \mathbb{E}_{\tilde{q}_\theta(\mathbf{x}_{<k})}\big[\sum_{x_k}\frac{e^{-\phi_\theta(x_k,\mathbf{x}_{<k})}}{\sum_{x_k}e^{-\phi_\theta(x_k,\mathbf{x}_{<k})}}\cdot\frac{\sum_{x_k}e^{-\phi_\theta(x_k,\mathbf{x}_{<k})}}{\mathbf{Z}_\theta}\frac{\partial}{\partial\theta}\phi_\theta(x_k,\mathbf{x}_{<k})\big]\\
&= \mathbb{E}_{\tilde{q}_\theta(\mathbf{x}_{<k})}\big[\sum_{x_k}\tilde{q}_\theta(x_k|\mathbf{x}_{<k})\mathbf{w}(\mathbf{x}_{<k})\frac{\partial}{\partial\theta}\phi_\theta(x_k,\mathbf{x}_{<k})\big]\\
&= \mathbb{E}_{\tilde{q}_\theta(\mathbf{x}_{<k})}\big[\mathbb{E}_{a\sim\tilde{q}_\theta(x_k|\mathbf{x}_{<k})}[\mathbf{w}(\mathbf{x}_{<k})\frac{\partial}{\partial\theta}\phi_\theta(x_k,\mathbf{x}_{<k})]\big]\\
&= \mathbb{E}_{x_k,\mathbf{x}_{<k}\sim\tilde{q}_\theta(x_k,\mathbf{x}_{<k})}\big[\mathbf{w}(\mathbf{x}_{<k})\frac{\partial}{\partial\theta}\phi_\theta(x_k,\mathbf{x}_{<k})\big]
\end{aligned}
$$
(18)

where $\mathbf{w}(\mathbf{x}_{<k})$ is also equal to $\frac{\sum_{x_k}e^{-\phi(x_k,\mathbf{x}_{<k})}}{\mathbf{Z}_\theta}$. Combining Eq. 16 and Eq. 18, we can obtain an equivalent form of the gradient of the negative phase without any expectation over $p_\theta$ as

$$
-\mathbb{E}_{\mathbf{x}_{<k}\sim\tilde{q}_\theta(\mathbf{x}_{<k})}[\mathbf{w}(\mathbf{x}_{<k})\frac{\partial}{\partial\theta}\log\tilde{q}_\theta(\mathbf{x}_{<k})] + \mathbb{E}_{x_k,\mathbf{x}_{<k}\sim\tilde{q}_\theta(x_k,\mathbf{x}_{<k})}[\mathbf{w}(\mathbf{x}_{<k})\frac{\partial}{\partial\theta}\phi_\theta(x_k,\mathbf{x}_{<k})], \quad (19)
$$

$$
\textbf{where}\quad \mathbf{w}(\mathbf{x}_{<k}) = \frac{\sum_{x_k}e^{-\phi(x_k,\mathbf{x}_{<k})}}{\mathbb{E}_{\mathbf{x}'_{<k}\sim\tilde{q}_\theta(\mathbf{x}_{<k})}[\sum_{x_k}e^{-\phi_\theta(x_k,\mathbf{x}'_{<k})}]}. \quad (20)
$$

## B THE FURTHER REFINEMENT OF $\mathbf{w}$

The reweighing weight $\mathbf{w}$ can be further deduced as

$$
\begin{aligned}
\mathbf{w}(\mathbf{x}_{<k}) &= \frac{\sum_{x_k}e^{-\phi(x_k,\mathbf{x}_{<k})}}{\mathbb{E}_{\mathbf{x}'_{<k}\sim\tilde{q}_\theta(\mathbf{x}_{<k})}[\sum_{x_k}e^{-\phi_\theta(x_k,\mathbf{x}'_{<k})}]} = \frac{\sum_{x_k}\frac{p_\theta(x_k,\mathbf{x}_{<k})}{\tilde{q}_\theta(\mathbf{x}_{<k})}}{\mathbb{E}_{\mathbf{x}'_{<k}\sim\tilde{q}_\theta(\mathbf{x}_{<k})}[\sum_{x_k}\frac{p_\theta(x_k,\mathbf{x}_{<k})}{\tilde{q}_\theta(\mathbf{x}_{<k})}]}\\
&= \frac{\frac{p_\theta(\mathbf{x}_{<k})}{\tilde{q}_\theta(\mathbf{x}_{<k})}}{\mathbb{E}_{\mathbf{x}'_{<k}\sim\tilde{q}_\theta(\mathbf{x}_{<k})}[\frac{p_\theta(\mathbf{x}_{<k})}{\tilde{q}_\theta(\mathbf{x}_{<k})}]} = \frac{\mu(\mathbf{x}_{<k})}{\mathbb{E}_{\mathbf{x}'_{<k}}\mu(\mathbf{x}_{<k})},
\end{aligned}
$$
(21)

where $\mu(\mathbf{x}_{<k})$ is defined as $\frac{p_\theta(\mathbf{x}_{<k})}{\tilde{q}_\theta(\mathbf{x}_{<k})}$.

## C  EXPERIMENTAL SETTINGS

In this section, we introduce the specific setup of different benchmarks in Table 6. We uniformly use Adam optimizer. The training will be stopped once the model has not obtained better performance for 20 epochs on the validation set. For translation tasks, the length of generated fake sentences, which is used for the computing of negative phase in Eq. 12, is dependent on the source sequence whilst for language modeling tasks, we fix the length of generated fake sentences as 50 during training. As for the model structures of the image generation task, we use the official structure reported by PixelCNN (van den Oord et al., 2016c) and Gated PixelCNN (van den Oord et al., 2016b) without modification. The source code will be released once upon acceptance. We use the same batch of samples generated autoregressively to approximate both the expectations in Eq.12 and weight $\mathbf{w}$ (*i.e.*, shared), which does not need to sample twice. The number of samples in a batch is dynamic while the maximum number of the total tokens in a batch are fixed (4096 in our experiments). If the length of sequences in a batch is 32, then it includes 4096 / 32 = 128 samples in total. It is a common strategy in language generation tasks, and has been used in many frameworks(e.g. Fairseq (Ott et al., 2019)). We generate samples autoregressively as many as the number of sequences in the current batch at each update iteration.

| Hyper-Parameters | IWSLT14 | WMT16 | | WiKiText103 | |
|---|---|---|---|---|---|
| | Tr-Base | Tr-Base | Tr-Large | Tr-Base | Tr-XL |
| Number of Layers | 12 | 12 | 12 | 6 | 16 |
| Hidden Embed Size | 512 | 512 | 1024 | 512 | 410 |
| FC-Layer Embed Size | 1024 | 2048 | 4096 | 2048 | 2100 |
| Attention Heads | 4 | 8 | 16 | 8 | 10 |
| Dropout | 0.3 | 0.3 | 0.3 | 0.1 | 0.1 |
| Learning Rate | 5e-4 | 1e-3 | 1e-3 | 5e-4 | 2.5e-4 |
| lr scheduler | inverse_sqrt | inverse_sqrt | inverse_sqrt | inverse_sqrt | cosine |
| Warm up Updates | 4000 | 4000 | 4000 | 4000 | 10000 |
| Weigth Decay | 1e-4 | 0.0 | 0.0 | 1e-2 | 0.0 |
| Coefficient $\lambda$ | 0.05 | 0.05 | 0.05 | 0.05 | 0.02 |
| E-ARM Start Epoch | 15 | 15 | 10 | 15 | 10 |

Table 6: Hyper-Parameters of different model structures and datasets. "Tr-Base", "Tr-Large", and "Tr-XL" indicate Transformer-Base, Transformer-Large, and Transformer-XL respectively

## D  MORE EXPERIMENTAL ANALYSIS

### D.1  EFFECT ON INCOHERENCE

In order to validate the effectiveness of our E-ARM for ameliorating the long-range coherence of generations, we undertake an experiment to assess the model's performance under different test sets with varying sentence lengths. We divided the test set of IWSLT14 (German → English, Italian → English, Spanish → English) translation dataset into three subsets ([0, 25], [25, 50], and [50, ∞)) based on the target sentence lengths. Then, we incrementally applied scheduled sampling technique and our E-ARM above the base transformer network, and tested their performances on these three subsets. Generally, the subset of samples with longer target sentences ([50, ∞)) should have been more affected by the long-range incoherence problem (lower BLEU score). In practice, we uniformly applied label smoothing and beam searching (with 5 beams) strategy for all experiments in Table 7.

Specifically, Table 7 shows that the base translation model improved performance for all three test sets with varying target sentence lengths after using the scheduled sampling technique, especially for the two sets [0, 25] and [25, 50) which had relatively short target sentence lengths (*e.g.* On German to English task, 38.20 - 37.72 = +0.48 points and 33.76 - 33.24 = + 0.52 points for [0, 25) and [25, 50) test sets respectively). We consider that this performance boost was achieved through alleviating the exposure bias problem, since scheduled sampling approaches (Ranzato et al., 2016; Zhang et al., 2019; Mihaylova & Martins, 2019) have been verified in mitigating the exposure bias problem. Besides, after applying our E-ARM together with the scheduled sampling technique, the

| Translation Task | Scheduled Sampling | E-ARM Training | Target Sentence Length | | | All Test |
|---|---|---|---|---|---|---|
| | | | [0, 25) | [25, 49) | [50, ∞) | |
| **De→En** | - | - | 37.72 ±0.04 | 33.24 ±0.06 | 30.86 ±0.07 | 34.61 ±0.08 |
| | ✔ | - | 38.20 ±0.07 | 33.76 ±0.03 | 31.08 ±0.06 | 35.10 ±0.04 |
| | ✔ | ✔ | 38.37 ±0.06 | 33.92 ±0.09 | 31.43 ±0.04 | 35.36 ±0.05 |
| **It→En** | - | - | 35.20 ±0.03 | 32.73 ±0.02 | 26.86 ±0.05 | 32.29 ±0.03 |
| | ✔ | - | 35.52 ±0.09 | 33.25 ±0.08 | 26.95 ±0.14 | 32.64 ±0.12 |
| | ✔ | ✔ | 35.56 ±0.10 | 33.33 ±0.13 | 27.21 ±0.07 | 32.82 ±0.11 |
| **Es→En** | - | - | 43.37 ±0.05 | 39.67 ±0.08 | 37.14 ±0.06 | 40.64 ±0.07 |
| | ✔ | - | 43.61 ±0.09 | 40.00 ±0.04 | 37.38 ±0.06 | 40.91 ±0.06 |
| | ✔ | ✔ | 43.84 ±0.10 | 40.35 ±0.05 | 38.07 ±0.04 | 41.58 ±0.07 |

Table 7: Performance comparison on the IWSLT14 test set with respect to the different lengths of sentences on three translation tasks (German to English, Italian to English, and Spanish to English). Performance is evaluated by BLEU score.

base model can further obtain additional performance gain. Specifically, the improvement on the longer sentence is more evident, since model can obtain large improvements on the [50, ∞) (*e.g.* On German to English task, 31.43 - 31.08 = +0.35 points for [50, ∞) test sets) than short sets [0, 25] and [25, 50] (*e.g.* On German to English task, 38.37 - 38.20 = +0.17 points and 33.92 - 33.76 = + 0.16 points for [0, 25) and [25, 50) test sets respectively). This phenomenon indicates that our E-ARM can resolve the incoherence problem to some extent.

## D.2 Effect on Exposure Bias

| Trans. Pairs | DE→EN | EN→DE | EN→IT | IT→EN | ES→EN | EN→ES |
|---|---|---|---|---|---|---|
| $\mathcal{N}$ | 14203 | 14554 | 14976 | 13952 | 16021 | 15359 |
| **Total** | 22148 | 23057 | 23654 | 23744 | 23860 | 22775 |
| **Ratio** | 64.12% | 63.12% | 63.31% | 59.76% | 68.33% | 67.43% |

Table 8: The effect of E-ARM on the exposure bias problem. Each test set of translation tasks contains 1K sentences selected randomly. $\mathcal{N}$ denote the ground truth words whose probabilities in the predicted distributions produced by E-ARM are greater than those produced by the baseline.

We follow the analytic experiments in the work (Zhang et al., 2019) to show that our E-ARM is capable of alleviating the exposure bias problem. Specifically, we randomly select 1K pairs from the training data for each translation pair and use the trained autoregressive model which applied E-ARM (Label Smoothing with smoothing factor 0.1 is applied during training while scheduled sampling is not used) to decode the source sentences, and then count the ground truth words whose probabilities in the predicted distributions produced by our E-ARM are greater than those produced by the baseline and denote the number as $\mathcal{N}$. The ratio of $\mathcal{N}$ to the total number of words tested is calculated. The detailed results are shown in Table 8. We find that the results on all different tasks are greater than 50%, which demonstrate the ability of our E-ARM in solving exposure bias problem.

## D.3 Analysis to model's Convergence

In this section, We will investigate the convergence of our E-ARM. To begin, we first train a base Transformer model ("Tr-Base" architecture shown in Table 6) on the IWSLT14 Spanish to English training set for baseline and E-ARM model respectively, and then record the training loss and test loss (in cross entropy) at the end of each epoch. The loss curves are plotted in the Figure 2. From Figure 2, we can see that (1) at the start of the training, our E-ARM converges slightly faster than the baseline. (2) As the training process progresses, the cross entropy of the baseline on the training set will gradually decrease, with a faster rate than E-ARM. On the other hand, the test loss curve of the baseline will fall at initially and then slowly rose after 50 epochs while E-ARM always remains stable convergence. This phenomenon also shows that our E-ARM model can effectively prevents over-fitting and produce better generalization.

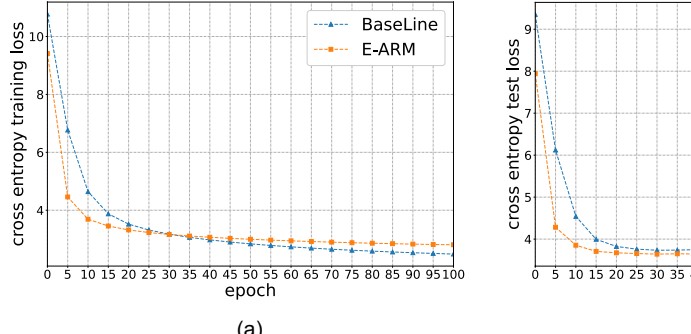

(a)                                    (b)

Figure 2: (a) Cross entropy loss curves on IWSLT14 Spanish to English translation task on training set. The blue and orange colors represent base model and E-ARM respectively; (b) Cross entropy loss curves on IWSLT14 Spanish → English translation task on test set.

## D.4 ANALYSIS TO TOP-K RE-SAMPLING

|  | Trans. Pairs | DE→EN | EN→DE | EN→IT | IT→EN | ES→EN | EN→ES |
|---|---|---|---|---|---|---|---|
| | 0 | 34.86 | 28.73 | 29.91 | 32.44 | 40.88 | 37.59 |
| $k$ | 5 | **34.93** | 28.85 | **30.04** | **32.56** | **41.01** | 37.66 |
| | 10 | 34.88 | **28.91** | 29.96 | 32.41 | 40.90 | **37.73** |

Table 9: The effect of Top-K correction in the inference stage. We tested BLEU scores of using different $k$ on different translation pairs of IWSLT14 dataset.

Top-K energy re-sampling in the inference stage is introduced by Bakhtin et al. (2021), which collects many candidate sequences generated autoregressively in the inference stage and then re-samples from them depending on their energy scores estimated by the network. To measure the contribution of the Top-K energy re-sampling in our method, we conduct ablation study to verify it by selecting different K = {0, 5, 10}. The results are shown in Table 9 by using BLEU score. From Table 9, we observe that the benefits brought by Top-K sampling is minor (K={5, 10}), when compared with model without Top-K sampling (K=0). These results also indicate that the performance improvements of our E-ARM are mainly from our joint-training, rather than Top-K energy re-sampling.

## D.5 EVALUATION WITH OTHER METRICS

| Trans. Pairs | Scheduled Sampling | E-ARM Training | Metrics | | | | |
|---|---|---|---|---|---|---|---|
| | | | ROUGE-1 ↑ | ROUGE-2↑ | ROUGE-L↑ | METEOR↑ | BLEU↑ |
| De → En | - | - | 66.51 | 43.69 | 63.69 | 64.35 | 34.61 |
| | ✔ | - | 66.83 | 44.08 | 64.02 | 64.61 | 35.10 |
| | ✔ | ✔ | **67.46** | **44.77** | **64.78** | **65.13** | **35.36** |
| It → En | - | - | 64.50 | 40.65 | 61.69 | 62.18 | 32.29 |
| | ✔ | - | 64.73 | 40.97 | 61.94 | 62.51 | 32.64 |
| | ✔ | ✔ | **65.27** | **41.51** | **62.49** | **62.80** | **32.82** |
| Es → En | - | - | 71.10 | 49.47 | 68.78 | 68.94 | 40.64 |
| | ✔ | - | 71.36 | 49.53 | 68.96 | 69.28 | 40.91 |
| | ✔ | ✔ | **71.91** | **50.17** | **69.65** | **69.63** | **41.58** |

Table 10: Comparison of ROUGE-1, ROUGE-2, ROUGE-L, METEOR, and BLEU scores between our approach E-ARM and the base ARGM trained just with cross-entropy loss on three translation pairs of IWSLT14 datasets. The value is expressed in percentage. We use "Tr-Base" as the network architecture.

To further evaluate the effectiveness of the our proposed E-ARM, we also evaluate our method by using other metrics, such as ROUGE Lin (2004) and METEOR Banerjee & Lavie (2005) for neural

machine translation. The results are shown in Table 10. In Table 10, the improvements of E-ARM in different metrics is consistent with the conclusion of Table 1, which further prove the effectiveness of our E-ARM model.

### D.6 EFFICIENCY STUDY

Our E-ARM has the advantage of being able to optimize an energy-based learning target using maximum log-likelihood, without the usage of MCMC procedures. The requirement to sample data from the autoregressive model at each update step, on the other hand, remains a possible element that could slow down the training process. Nonetheless, the extra overheads are still acceptable when compared to sampling data using MCMC algorithms. The reasons are provided in below:

Assuming that the forward processes of the Transformer, with a length $n$ sentence as the input, have the time cost $\tau$. We tested the time cost of gradients back-propagation for Transformer on Tesla V100 GPU. We found that the time cost of the backward process is approximately twice as the forward process, which is marked as $2\tau$. Therefore, the time cost of one step update is approximate $3\tau$. Autoregressively generating a sequence of length $n$ by Transformer necessitates $n$ feedforward processes, as each predicted token must use all previously created tokens as input. One fact is that we simply need to use the previously produced $k$ tokens as the input at each time step $k$, and gradient back-propagation is not required during the generation.

As a result, the time cost of generating a fake sentence for energy-based training in Eq.12 is actually $\frac{1}{n}\tau + \frac{2}{n} + \cdots + \frac{n-1}{n}\tau = \frac{n-1}{2}\tau$. Considering the IWSLT14 German to English translation task, which has training samples with length 20 in average, the time cost of generating fake data in each iteration is roundly $9.5\tau$. Furthermore, the generated fake sentence will be fed into the transformer and included in the overall loss computation, resulting in an extra forward and backward proce-

| Model | S.S. | w/E-ARM | Sec./100 iter. |
|---|---|---|---|
| **Tr-Base** | - | - | 27.3 |
| | ✔ | - | 30.1 |
| | - | **Autoreg.** | 145.8 |
| | ✔ | **Autoreg.** | 149.2 |
| | - | **20 steps SGLD** | 630.6 |
| | - | **50 steps SGLD** | 1452.3 |

Table 11: Efficiency performance on IWSLT14 German→ English, evaluated with BLEU. We uniformly use 12 layer "Tr-Base" in Table 6. "**S.S.**" denotes Scheduled Sampling."Autoreg." indicates optimizing E-ARM with Eq.12 by sampling fake data from autoregressive models. "* steps SGLD" represents optimizing our E-ARM with Eq.6, the fake data is sampled at the first transformer layer's output by SGLD with * steps.

dure apart from the update of original input. Thus, the total time cost of our E-ARM's one update is $15.5\tau$, which is about 5.2 times as great as vanilla training. Table 11 shows the time cost of training a 12 layer transformer with 100 iterations. The time cost of our E-ARM roughly coincides the extra time cost as we analyzed above. For long sequence tasks like image generation and language modeling, which usually have sequences consisting of hundreds of tokens, we randomly truncate a continuous sequence with length 50 for energy-based training in Eq.12.

When it comes to the MCMC sampling, one problematic issue is that for sequential data like text, the intrinsic discrete property prevents it from applying MCMC in the data space, which forces us to apply it in the latent feature space. Here, we take the SGLD (Welling & Teh, 2011b) algorithm for example. Assuming that we apply the SGLD at the first layer of the network, then the time cost of one SGLD iteration is about $3\tau$ either. Since the SGLD process requires $k$ iterations to reach convergence, the total time cost of one update of our E-ARM with MCMC process is $(3k + 6)\tau$. In practice, the $k$ is usually set as 100 for stable training Grathwohl et al. (2020), which results in the time cost being $\frac{(3\times100+6)\tau}{3\tau} = 102$ times as large as the vanilla training. For short-run SGLD, which takes $k$ as 20 with a sacrifice of performance, it still leads to the time cost being 22 times as large as the vanilla training.

## D.7 CASES STUDIES

To better understand the advantages of our method in correcting error tokens, we also prepare some translation cases in IWSLT14 German → English, as shown in Table 12.

| Source Sentence(German) | Predicted Target Sentence(English) |
|---|---|
| wenn ich ihnen 600 zeitschriften zeige und sie in 10 kategorien aufteile oder ich ihnen 400 zeitschriften zeige, und diese in 20 kategorien aufteile, dann glauben sie, dass ich ihnen mehr auswahl und eine bessere auswahlerfahrung gegeben habe, als ich ihnen die 400 gegeben hätte gegenüber dem, wenn ich ihnen die 600 gegeben hätte. | **GroundTruth**: if i show you 600 magazines and i divide them up into 10 categories, versus i show you 400 magazines and divide them up into 20 categories, you believe that i have given you more choice and a better choosing experience if i gave you the 400 than if i gave you the 600.

**Baseline**: if i show you 600 magazines and i split them in 10 categories, or i'm showing them 400 magazines, and i'm going to split them up into 20 categories, you think i've given them more choices and better choice than i would have given them the 400 over the time that i gave them the 600.
**Baseline + S.S.**: if i show you 600 magazines and i give you 400 magazines in 10 categories, and i give you 400 magazines, and i can split them up in 20 categories, then you think i've given you more choice and a better selection than i would have given you the 400 of which if i gave you the 600.
**Ours**: if i show you 600 magazines and i divide them into 10 categories, or i show you 400 magazines, and i divide them into 20 categories, you think i've given you more choices and better selection experience than i gave you the 400 of whom if i gave you the 600. |
| und ich weiß definitiv, dass es für mich – in meiner situation – sehr gefährlich wäre, anzufangen, diesen dunklen pfad der vermutung sozusagen herunterzu-sickern – besonders in dem umstand, in dem ich mich in meiner karriere gerade befinde. | **GroundTruth**: and i definitely know that, in my case – in my situation – it would be very dangerous for me to start sort of leaking down that dark path of assumption, particularly given the circumstance that i'm in right now in my career.
**Baseline**: and i know definitely, for me, it would be very dangerous to begin to do this dark path of suspect – especially in the circumstance that i'm in my career right now.
**Baseline + S.S.**: and i know definitely it would be – in my situation – very dangerous to start, to kind of settle down this dark path of presumption – especially in the circumstance in which i'm in my career right now.
**Ours**: and i definitely know that it's for me – in my situation – very danger-ous to start to sickle down this dark path of suspicion, in particular, in the circumstance of where i'm in my career right now. |
| wir haben das licht ausgeschaltet, legten es in ein vakuum und saugten die ganze luft aus und kühlten es bis fast zum jetzt, ganz alleine im aufzug, war das stück metall frei, sich zu verhalten wie immer es wollte. | **GroundTruth**: we turned off the lights, and then we put it in a vacuum and sucked out all the air, and then we cooled it down now, all alone in the elevator, the little chunk of metal is free to act however it wanted.
**Baseline**: we turned the light off, put it in a vacuum and sucked it out all the air and cooled it up until almost now, all the way alone, the piece of metal was open to behave as it was.
**Baseline + S.S.**: we turned the lights off, we put it into a vacuum, and we sucked all the air, and we cooled it all the way up to now, all over the place, the piece of metal was free to behave whatever it wanted.
**Ours**: we turned off the lights, we put it into a vacuum and we sucked all the air out, and we cooled it up until almost now, all alone in the elevator, the piece of metal was free to behave whatever it wanted. |
| und im grunde können sie das betrachten, wissen sie, als eine tyrannei des erin-nernden selbst, und sie können sich das erinnernde selbst denken als eins, das sozusagen das erlebende selbst schleppt durch erfahrungen, die das erlebende selbst nicht braucht. | **GroundTruth**: and basically you can look at this, you know, as a tyranny of the remembering self, and you can think of the remembering self sort of dragging the experiencing self through experiences that the experiencing self doesn't need.
**Baseline**: and basically, you can think of this, you know, as a tyranny of self, and you can think of the memorable self as one that kind of weaves the living self through experiences that don't need the life itself.
**Baseline + S.S.**: and basically, you can look at this, you know, as a tyrannei of memorial self, and you can think of the memorial self as one that kind of sucks the living self through experiences that don't need the living self.
**Ours**: and basically, you can look at that, you know, as a tyranny of the re-membering self, and you can think of the memory itself as one, which is sort of dragging the living self through experiences that the living self doesn't need. |
| wir sind an der schwelle zu erstaunlichen, erstaunlichen ereignissen auf vielen gebieten. und doch denke ich wirklich, dass wir hunderte, 300 jahre vor die aufklärung zurück gehen müssten, um eine zeit zu finden, in der wir fortschritt bekämpft haben, in der wir über diese dinge heftiger getritten haben, an mehr fronten als jetzt. | **GroundTruth**: we're on the verge of amazing, amazing events in many fields, and yet i actually think we'd have to go back hundreds, 300 years, before the enlightenment, to find a time when we battled progress, when we fought about these things more vigorously, on more fronts, than we do now.

**Baseline**: we are at the threshold of amazing, amazing events in many areas, and yet i really think that we have to go back hundreds and 300 years before the enlightenment to find a time when we have fought progress in which we have driven more of these things than now.
**Baseline + S.S.**: we're at the threshold of amazing, amazing events in many areas. and yet, i really think that we have to go back hundreds and hundreds of years before the enlightenment to find a time when we have struggled with progress in which we have driven on these things more powerful, more fronts than now.
**Ours**: we're at the threshold to amazing, amazing events in many areas, and yet i really think that we have to go back hundreds and 300 years before the en-lightenment to find a time when we fought progress, where we've been fighting about these things to more fronts than we have now. |

Table 12: Translation cases on IWSLT14 De→En test set, generated by the baseline method, baseline with scheduled sampling and our E-ARM. The italic font means the mismatch translation

# E   MORE DISCUSSION OF RELATED WORKS

The seminal idea of combing a generative model and an energy-based model has been explored by a plethora of great works (Pang et al., 2020; Durkan & Nash, 2019; Xie et al., 2019; 2020; Xiao et al., 2021; Bakhtin et al., 2021). Our E-ARM can be considered as a member of this family of models in general, but it has a different mechanism and goal than the others. In particular, Pang et al. (2020) aimed to learn an energy-based model (EBM) in the latent space of a generator model, so that the EBM can act as a prior model on the generator model's top-down network. They believe that the energy-based correction of the prior noise distribution will benefit the subsequent generator's generating process. Furthermore, Xie et al. (2019) attempted to learn the conditional distribution of a high-dimensional output given an input by combining the efforts of a fast thinking initializer, which generates the output and a latent vector, and a slow thinking solver, which learns an objective function in the form of a conditional energy function, so that the output can be generated by optimizing the objective function, or more rigorously by sampling from the conditional energy-based model. A similar work is GAMs (Parshakova et al., 2019a;b; Khalifa et al., 2021), which combine an autoregressive component with a log-linear component, allowing the use of global a priori features to compensate for lack of data. Moreover, VAEBM, a symbiotic composition of a variational auto-encoder and an EBM, was proposed by (Xiao et al., 2021). It can use a state-of-the-art VAE to capture the general mode structure of the data distribution while relying on its EBM component to explicitly eliminate non-data-like regions from the model and refine the generation samples. In addition, Bakhtin et al. (2021) designed a novel mechanism to train an unnormalized energy-based models for modeling joint sequence by working in the residual of a pretrained locally normalized language model and training using noise contrastive estimation. All of the above models require an additional network to learn the energy scores, which prevents the base autoregressive model from benefiting from EBM's properties in modeling the joint distribution in a more temporally coherent manner. In contrast, by carefully constructing an energy-based learning objective and its corresponding optimization procedure, we are able to smoothly integrate energy surface learning into autoregressive networks that do not require additional learnable parameters. Rather than proposing a new generative model, our method is more likely to a novel training pattern for training a better autoregressive model. Recently, instead of constructing an autoregressive model in the data space, Xu et al. (2021b) have proposed a unique way which uses autoregressive models in the latent space followed by a decoder which decodes the autoregressively generated latent feature into the original data space. They attempt to learn a structured representation space where dimensions are ordered based on importance and trade off the sample quality for computational efficiency by truncating the dimensions of latent generations. Their work is orthogonal to ours. We think the combination between our E-ARM with anytime sampling is also a valuable work which is worth exploration in the future.

