# OpenReview forum: "YOUR AUTOREGRESSIVE GENERATIVE MODEL CAN BE BETTER IF YOU TREAT IT AS AN ENERGY-BASED ONE"
_ICLR.cc/2022/Conference — ICLR 2022 Submitted_

### Official Review · Reviewer_SjXn · 2021-10-31

**Correctness:** 3
**Technical Novelty And Significance:** 2
**Empirical Novelty And Significance:** 3
**Recommendation:** 5
**Confidence:** 2

**Main Review:**

 # Strong points

- The considered problem is important, and the authors' novel framing of it is well-motivated: using an EBM model that takes the global information into account to mitigate the gap between training and inference.

- A simple algorithm is derived under the defined joint distribution. They use the model distribution as the importance distribution, and It's faster to estimate the weights in the importance sampling than MCMC procedures in traditional EBM.

- Experiments are designed appropriately to showcase the resulting behavior of the model. The proposed method outperforms the base autoregressive models, and is compatible with other techniques and architectures.


# Weak points

- The design of the joint distribution seems a bit arbitrary. Could the author better explain the rationale behind the design?  For example, there is large flexibility for $\phi$ in Eq (4). If $\phi=-\log q(x_{\le k})$, then the importance weight is $w(x_{<k}) = \frac{q(x_{< k})}{E[q(x_{< k})]}$. It seems to me that this choice is a more intuitive one.  In addition, the joint distribution is essentially defined by conditional distributions. It seems to me that ``we compel the ARGM to fit not only the conditional distributions but also the joint distribution at each time step" overclaims the actual modeling of the joint distribution.

- Lack of comparison/discussion of relevant works. For exposure bias:  [1] adopt a discriminator as the energy-based model to correct the ancestral sampling in every step. For long-range dependency: [2] instead learns a global ordered representation, like PCA, and trains autoregressive models on top of the representation. ([2] would still incur exposure bias). The paper also does not compare to those ``two stages" methods in intro.

# Minors

- It seems that there should be a $\sum_k$ in the RHS of Eq. (5).

[1] Yuntian Deng and Anton Bakhtin and Myle Ott and Arthur D. Szlam, Residual Energy-Based Models for Text Generation, abs/2004.11714

[2] Yilun Xu and Yang Song and Sahaj Garg and Linyuan Gong and Rui Shu and Aditya Grover and Stefano Ermon, Anytime Sampling for Autoregressive Models via Ordered Autoencoding, abs/2102.11495

**Summary Of The Paper:**

This paper considers the problem of exposure bias and lack of long-range coherence in auto-regressive models. Sampling in such models is sequential, so the error would propagate through ancestral sampling. The authors propose to combine autoregressive models with EBM, which does not require the assumption of distribution factorization. Existing works use a two-stage optimization, where an EBM is trained on top of the auto-regressive models. The authors instead jointly minimize an energy-based learning objective and the vanilla MLE. They also present a way of estimating the gradient of negative samples through importance sampling. Extensive experiments on NLP and vision tasks show that the proposed learning objective reduces the discrepancy between training and inference.

**Summary Of The Review:**

There are some loose threads have been left by the authors. I would increase the scores if the authors address some of the points above.

---

> ### Author Response · Authors · 2021-11-22
> **Response to reviewer SjXn, about Q2-Q3**
>
> Thanks for your valuable comments. Below are our responses to your comments:
>
> **[Q2 Lack comparison of relevant works]**
>
> Thanks you for suggestions. We have noticed some related works, and the below are some results of comparisons:
>
> - We currently conduct experiments to compare our E-ARM with Residual EBMs [1]. Considering the limitation of computing resources and time, the experiments of our E-ARM performance on the two benchmarks (CC-News, Toronto books) reported by [1] is still undergoing, since [1] has applied 8 DGX nodes and each with 8 Nvidia V100s. We will report the results on these two datasets when it is finished in the future. To make a fast verification, we have implemented their Residual EBMs on the Wikitext103 dataset based on transformer-xl network and achieved 23.96 PPL, which is slightly worse than the performance 23.81 PPL of our E-ARM with transformer-xl. This results can show the effectiveness of our E-ARM is comparable with Residual EBMs. Besides, the Residual EBMs requires an extra energy estimation network which introduced the same learnable parameters as the base autoregressive networks, while our method only uses single autoregressive network (i.e., their model requires 2 times parameters as ours). We will updated the results once the rest experiments are finished.
>
> However, our method is more likely to a novel training pattern for training a better autoregressive model to solve the exposure bias and incoherence problems, without introducing any further learnable parameters or special architectures. As a result, the comparison with methods which solve exposure bias, like scheduled sampling listed in our paper, is more reasonable compared to those methods, which proposed a new generative model (VAEBM [3], etc).
>
> On the other hand, the Anytime Sampling [2] method is an excellent work, which uses autoregressive models in the latent space followed by a decoder which decodes the autoregressively generated latent feature into the original data space. They attempt to learn a structured representation space where dimensions are ordered based on importance and trade-off the sample quality for computational efficiency by truncating the dimensions of latent generations. We have included the citation and discussion of this paper in our latest version, however, we argue that this work is orthogonal to our method. First, they focus on improving the efficiency of autoregressive models while we pay attention to improving the performance of the autoregressive network. Second, their novel mechanism is compatible with ours, they transplant autoregressive models into the latent feature level while our E-ARM adheres to autoregressive model and can also be transplanted into the latent space in their manner. Nevertheless, exploring an effective way to combine our E-ARM and this anytime sampling, in my opinion, is an interesting and promising direction to further improve the performance of autoregressive models, which we might research in the future.
>
> **[Q3 $\sum_k$ in the RHS of Eq. (5)]**
>
> Thanks for pointing out this mistake. We have fixed this part in the latest version.
>
> [1] Yuntian Deng, Anton Bakhtin, Myle Ott, Arthur Szlam, and Marc’Aurelio Ranzato. Residual energy-based models for text generation
>
> [2] Yilun Xu, Yang Song, Sahaj Garg, Linyuan Gong, Rui Shu, Aditya Grover, and Stefano Ermon. Anytime sampling for autoregressive models via ordered autoencoding.
>
> [3] Zhisheng Xiao, Karsten Kreis, Jan Kautz, and Arash Vahdat. VAEBM: A symbiosis between variational autoencoders and energy-based models.

---

> > ### Comment · Reviewer_SjXn · 2021-12-01
> > **Thanks for the response**
> >
> > I thank the authors for the detailed response. It's good to hear that E-ARM outperforms Residual EBMs in the initial experiment.
> >
> > Unfortunately, I do not immediately see the problem of $-\log q(x_{\le k})$. For the first concern, the multiplication can be easily paralleled. Indeed, the same amount of computation is needed in the first term of Eq. (5). One can also use the $\tilde{q}$ trick to limit the time step. I don't understand the second concern. Why will it be challenging? What's the difference between using conditional distributions and the logits here?

---

> > > ### Author Response · Authors · 2021-12-07
> > > **Response to Reviewer SjXn for the concern of energy function's definition.**
> > >
> > > Thanks for your response, we are sorry that the former explanation about the concern of the definition of energy function $\phi$  might be unclear and somewhat misleading. To make a clear understanding, we explain the potential problem of setting $\phi$ as $-\log q_\theta(x_{\leq k})$  in this response more concretely.
> > >
> > > Specifically, if we set $\phi$ as $-\log q_\theta(x_k, x_{<k})$, the definition of $p_\theta(x_k, x_{<k})$ would become as
> > >
> > > $$p_\theta(x_k, x_{<k}) = q_\theta(x_{<k}) \cdot q_\theta(x_{\leq k}) \cdot \frac{1}{Z} = \frac{q_\theta(x_{<k}) \cdot q_\theta(x_{\leq k})}{\sum_{x_k} \sum_{x_{<k}}q_\theta(x_{<k}) \cdot q_\theta(x_{\leq k})} = \frac{q_\theta(x_{< k}) \cdot q_\theta(x_{< k}) q_\theta(x_k | x_{< k})}{\sum_{x_k} \sum_{x_{<k}}q_\theta(x_{< k}) \cdot q_\theta(x_{< k}) q_\theta(x_k | x_{< k})} = \frac{q_\theta(x_{< k}) \cdot q_\theta(x_{< k}) q_\theta(x_k | x_{< k})}{\sum_{x_{<k}}q_\theta(x_{< k}) \cdot q_\theta(x_{< k}) } = \frac{q_\theta(x_{<k})}{E_{x_{<k} \sim q_\theta(x_{<k})}[q_\theta(x_{<k})] } \cdot q_\theta(x_k, x_{<k}),$$
> > >
> > > which indicates that $p_\theta(x_k, x_{<k}) = \frac{q_\theta(x_{<k})}{E_{x_{<k} \sim q_\theta(x_{<k})}[q_\theta(x_{<k})] } \cdot q_\theta(x_k, x_{<k})$. However, such a formulation is problematic under the learning objective of Eq.5. The reason is that in the learning objective of Eq. 5, the re-weighted joint distribution $p_\theta(x_k, x_{<k})$ has been required to approach the real distribution $p_d(x_k, x_{<k})$, and at the same time, the objective to make conditional distribution $q_\theta(x_k | x_{<k})$ approach $p_d(x_k|x_{<k})$ at each time step would result in the joint distribution $q_\theta(x_k, x_{<k})$ to fit the real distribution $p_d(x_k, x_{<k})$. In other words, we need $q_\theta(x_k, x_{<k})$ and $p_\theta(x_k, x_{<k})$ to approach the real distribution $p_d(x_k, x_{<k})$ in parallel, while it is difficult to achieve since $q_\theta(x_k, x_{<k})$ would be almost impossible to be same as $p_d(x_k, x_{<k})$ for all possible sequence ($x_k, x_{<k}$) if we set $\phi$ as $-\log q_\theta(x_{\leq k})$  (the weight term $\frac{q_\theta(x_{<k})}{E_{x_{<k} \sim q_\theta(x_{<k})}[q_\theta(x_{<k})] }$  can hardly be 1 for all $x_{<k}$, which can only happen for a uniform distribution).
> > >
> > > In contrast, our definition of the energy function is feasible for the optimization. The formulation of our definition of the product EBM can be written as (considering that $q_\theta(x_k|x_{<k}) = \frac{e^{-\phi_\theta(x_k, x_{<k})}}{\sum_{x_{k}} e^{-\phi_\theta(x_k, x_{<k})}}$)
> > >
> > > $$p_\theta(x_k, x_{<k}) = q_\theta(x_{<k}) \cdot \frac{e^{-\phi_\theta(x_k, x<k)}}{Z_\theta}=q_\theta(x_{<k}) \cdot q_\theta(x_k|x_{<k}) \cdot \frac{\sum_{x_{k}} e^{-\phi_\theta(x_k, x_{<k})}}{E_{x_{<k}\sim q_\theta(x_{<k})}[\sum_{x_{k}} e^{-\phi_\theta(x_k, x_{<k})}]} = q_\theta(x_k, x_{<k}) \cdot \frac{\sum_{x_{k}} e^{-\phi_\theta(x_k, x_{<k})}}{E_{x_{<k}\sim q_\theta(x_{<k})}[\sum_{x_{k}} e^{-\phi_\theta(x_k, x_{<k})}]},$$
> > >
> > > which allows both the joint distribution $q_\theta$ modeled by the autoregressive model and the joint distribution $p_\theta$ modeled by the introduced product EBM to achieve the real distribution $p_d$ simultaneously, since the summation of logits' exponentials over all possible tokens at time step k can be same among all possible sequences ($x_k, x_{<k}$) (the softmax operation is insensitive towards the magnitude of logits, for example, the softmax result of the vector [0.1, 0.3, 0.6] are equal to the softmax result of the vector [0.1 +C, 0.3 + C, 0.6+C]. As a result, it assigns an extra degree of freedom which make the optimization feasible. In contrast, the weight term $\frac{q_\theta(x_{<k})}{E_{x_{<k} \sim q_\theta(x_{<k})}[q_\theta(x_{<k})] }$  can not get such a degree of freedom to allow the optimization goes on since $q_\theta(x_{<k})$ is confined to its original learning objective  $\min KL(p_d(x_{<k}) || q_\theta(x_{<k}))$ ).
> > >
> > > However, although we think the definition of $\phi=-\log q_\theta(x_{\leq k})$ is infeasible, we do not deny that there might be some other formulation of $\phi$ which makes sense, e.g. from a technical perspective, using an extra network to estimate the $\phi$ is also a plausible way, but it disobeys the initial motivation of our method— improving the autoregressive model without any extra parameters and special architectures.

---

> ### Author Response · Authors · 2021-11-22
> **Response to reviewer SjXn, about Q1**
>
> Thanks for your valuable comments. Below are our responses to your comments:
>
> **[Q1 The rationale behind the design]**
>
> It seems the question is about the rationale behind the selection of energy function $\phi$ in Section 3. The basic motivation of our method is that the value of pre-softmax activation (logit) can be utilized to measure the quality of generated sequences, since it can serve as an extra degree of freedom. The underlying reason for such design is that if we only use the logit to calculate the softmax, the magnitude of the logit will be meaningless since the softmax operation are senseless to the magnitudes (the softmax result of the vector [0.1, 0.3, 0.6] are equal to the softmax result of the vector [0.1 +C, 0.3 + C, 0.6+C]. Here the softmax operation is insensitive to the magnitude of logits' value, which can be utilized in modeling). As for the alternative energy function $\phi = - \log q(\textbf{x}_\{\leq k\})$
>
> mentioned in the question, since we designed our definition of $\phi$ out of the intuition discussed above, we did not consider other formulations of $\phi$. However, we suppose there are several flaws of such definition without strict proof. First, the computing of $q(\textbf{x}_{\leq k})$ is a problematic issue since it requires a multiplication over the conditional distributions with all different time steps $\leq k$. Second, since
>
> $q(\textbf{x}_{\leq k})=\prod q(x_i|\textbf{x}_\{<i\})$,
>
> each value of  $q(x_i|\textbf{x}_{<i})$ has its corresponding meaning, which is the conditional probability of tokens given a context. Thus, forcing the product of them to be further involve into the learning of the energy-based objective will be challenging.
>
> Moreover, we know that the joint distribution is essentially defined by conditional distributions. Purely optimizing the KL-divergence between the real conditional distribution and the modeled conditional distributions in the vanilla training, can not ensure a desirable joint distribution modeled by ARGMs, since in practice teacher-forcing, which uses the ground truth as the input context, is usually adopted for more stable convergence. This training manner will result in the exposure bias problem(See section 2.3 in our paper). In contrast, we use EBM to append a constrain to align the joint distribution of autoregressively generated sequences and the real one, which can mitigate the aforementioned problem to some extent.  We admit that such a statement "we compel the ARGM to fit not only the conditional distributions but also the joint distribution at each time step"  might be an overclaim, our actual meaning is that ``we attempt to reduce the intrinsic problems of autoregressive models, such as exposure bias and weak temporal coherence, by optimizing an energy-based learning objective, which uses samples autoregressively generated''. We have modified this part in the latest version.

---

### Official Review · Reviewer_DZsJ · 2021-11-01

**Correctness:** 3
**Technical Novelty And Significance:** 3
**Empirical Novelty And Significance:** 3
**Recommendation:** 6
**Confidence:** 4

**Main Review:**

The approach is interesting and new (to the best of my knowledge). It is motivated by the exposure bias or in other terms *"a discrepancy of the input context distributions between the training and inference stages"*, which ARGM are reported to suffer from.
Accordingly, the experiments show improved performance when combining several ARGM with the proposed E-ARM method.

That alone, however, does not yet make strong paper (rating 8 or higher) in my opinion, because the authors don't study the effect of their method in detail. Instead, they confine themselves to just reporting improved performance on selected, and not exactly the latest benchmarks (e.g. PixelCNN is from 2016).
What I would wish in order to increase my rating are
* experiments that reveal when and how exposure bias is actually a problem (e.g. depending on the model expressiveness)
* experiments that show that this bias is indeed decreased when using E-ARM
* comprehensive studies on the cost this decreased bias (=improved performance) comes with

Being the devil's advocate, my suspicion could be that the improved performance is not actually due to $p_\theta$ (Eq. 4) but due to the heavily truncated back propagation in the second term of (5). A small footnote tells that BP is just two time steps back. The fact that this seems to be crucial for the method requires further investigation, and would make the paper much stronger. Besides, it would be easy to check by just using $\tilde q_\theta$ instead of $p_\theta$ in (5).

The paper was ok to follow, but can still be improved in writing in the sense that the mathematical derivations are sometimes hard to follow.
A lot of the separated equations contain just standalone terms instead of equations. Their references and the relation to them have to be found in the text, which hinders readability.

Eq. 5 is misleading, although the intention of it might be clear. The optimisation is over a sum over all $k$  and under the expectation over $\mathbf x\sim p_d$. That should be made explicit for better understanding.

The authors mention density estimation a couple of times, although the proposed method is clearly dealing with discrete distributions. In particular (footnote 1 would be wrong for densities).

Sec. 2.1, first line: $\mathbf x\in\mathbb R^K$, to be consistent with what follows

S2.3: *"Besides,autoregressive decoding typically greedily selects the most probable token at each time step, given
the ones previously selected.”*
I would disagree here. What you usually do is sampling the next token. That greedily decoding to the conditional mode is far from being ideal  is well known, and is by no means a shortcoming of autoregressive models.


**Summary Of The Paper:**

The authors propose an approach to integrate energy based models (EBM) into into general auto-regressive generative models ARGM. They propose to combine the negative log-likelihood loss of auto-regressive models with a second energy based loss, which serves as a kind of regularisation loss (steered with a hyper-parameter lambda). They claim that this approach mitigates the Exposure Bias which is frequently found in ARGM. A learning algorithm based on Hinton's wake-sleep algorithm is derived that approximates the EBM part of the loss with importance sampling thus avoiding expensive MCMC approximations.
Experiments form the NLP and the image generation domain are presented that show outperformance over selected baselines.


**Summary Of The Review:**

An interesting idea that can lead to improved performance on certain benchmarks. However, deeper insights about the effect that the model addresses (exposure bias mitigation) as well as estimations of the increased overhead are missing in the experiments, thus limiting the scope of the paper.

---

> ### Author Response · Authors · 2021-11-22
> **Response to reviewer DZsj, about Q3-Q6**
>
> **[Q3 Writing Improvement]**
>
> Thanks for your suggestions. We have improved the writing of our paper in the latest manuscript, and we will continue to improve our paper to make a more clear description in the final version.
>
> **[Q4 Eq.5 is misleading]**
>
> Thanks for pointing out this issue, we have updated Eq.5 in the latest version. The initial version of Eq.5 only considers single time step case, and we have updated it into a summation over all time steps k to make a clear description.
>
> **[Q5 Consistent for the first line of Sec. 2.1]**
>
> Thanks for your suggestions, we have fixed this typo in the latest manuscript.
>
> **[Q6 Greedily Decoding]**
>
> Thanks for your suggestions. We agree that our statement is a bit prejudice. The original intention in this part is to state that greedy search (including beam search) only considers local optimum. We have adjusted the description of this part in the latest version.

---

> ### Author Response · Authors · 2021-11-22
> **Response to reviewer DZsj, about Q1-Q2**
>
> Thanks for your valuable comments. Below are our responses to your comments:
>
> **[Q1 Experiments to reveal exposure bias/bias is indeed decreased/the cost the decreased bias comes with]**
>
> Thanks for your advice. Exposure bias is usually a common problem [1][2] in autoregressive models (e.g., language generation, text translation, etc). This problem usually is caused because the previously generated tokens have incorrect tokens, and thus produce error propagation to affect the subsequent generation. Below are our answers to address your concerns:
> 1.  To validate this, a simple experiment to validate exposure bias is to test the results of longer sequence generation. Therefore, we split datasets as multiple subsets by sequence length and the corresponding results are reported in Table 7 (please see Appendix D.1 in the latest manuscript). In Table 7, for example, on German $\rightarrow$ English task, the bleu score obtains 37.72 at the subset [0, 25), while only achieves 30.86 at the subset [50, $\infty$). This phenomenon manifests that the existence of exposure bias in sequence generation, especially in the longer sentence;
> 2. To validate that this bias is indeed decreased when using E-ARM, we follow previous experiences [1] to validate the ability of the proposed method in solving exposure bias, that is counting the ground truth words whose probability in the predicted distributions produced by E-ARm are greater than those produced by the baseline. The results are as follow (More details refer to Appendix D.2):
> | Trans. Pairs | DE → EN | EN → DE | EN → IT | IT→EN | ES→EN  | EN→ES |
> | --- | --- | --- | --- | --- | --- | --- |
> | N | 14203 | 14554 | 14976 | 13952 | 16021 | 15359 |
> | Total | 22148 | 23057 | 23654 | 23744 | 23860 | 22775 |
> | Ratio | 64.12% | 63.12% | 63.31% | 59.76% | 68.33% | 67.43% |
>
> We find that the results on different tasks are nearly 60~68% (greater than 50%), which demonstrates our E-ARM can reduce bias indeed;
>
> 3.  In addition to the above two experiments, to better demonstrate the cost the decreased bias comes with, we first evaluate disabling Top-K re-sampling to validate that the performance improvements are mainly from our joint-training. The results are shown in Table 9 (in appendix D.4), and we find that Top-K re-sampling can only improve the model marginally (at most 0.18 points), which can be neglected. Hence, these results also indicate the improvements are mainly from the joint training in our E-ARM. Besides, we also investigate the training efficiency between ours and MCMC sampling in Appendix D.6 and some cases studies as the demos for reference in Appendix D.7.
>
> **[Q2 The improved performance is not actually due to $p_\theta$ (Eq. 4) from the truncated back-propagation in the second term of Eq. 5]**
>
> Well, we have actually tried the untruncated version of $\tilde{q_\theta}$ to define $p_\theta$ in practice. However, the training can hardly tend to convergence. The reason for that we have analyzed in section 4 of our paper. Specifically, if we use the untruncated version of $\tilde{q_\theta}$ (which we denotes as $q_\theta$ hereafter), since the gradient $\nabla_\theta \log \tilde{q_\theta} (\textbf{x}_{<k})$
>
> ( $\nabla_\theta \log q_\theta (\textbf{x}_{<k})$  for untruncated version)
>
> in the first part of Eq.9 can be decomposed into the summation of several gradients of conditional distributions $ \sum_{l=1}^{k} \nabla_\theta \log q_\theta (x_l |\textbf{x}_{<l} ) $
>
> which results in model get stronger training signals for those earlier time steps (e.g. the model will get strongest training signals on the first time step, since $\sum_{l=1}^{k}\nabla_\theta \log q_\theta(x_l |\textbf{x}_{<l} )$ includes the first time step (l=1), no matter which specific time step k it is).
>
> Besides, "it would be easy to check by just using $\tilde{q_\theta}$ instead of $p_\theta$ in (5)" is not feasible in fact, since  $\tilde{q_\theta}$ is essentially a copy of the autoregressive model $q_\theta$ with those conditionals of long-range time steps fixed. If we used  $\tilde{q_\theta}$ instead of $p_\theta$ in (5), there would be no difference between the first KL-divergence and the second KL-divergence, since the joint  $\tilde{q_\theta}$ can be decomposed into a product of conditionals.
>
> [1] Wen Zhang, Yang Feng, Fandong Meng, Di You, Qun Liu. Bridging the Gap between Training and Inference for Neural Machine Translation.
>
> [2] Samy Bengio, Oriol Vinyals, Navdeep Jaitly, Noam Shazeer. Scheduled Sampling for Sequence Prediction with Recurrent Neural Networks.

---

> > ### Comment · Reviewer_DZsJ · 2021-12-01
> > **thanks for your response**
> >
> > I had a look at the additional experiments in the appendix. While they demonstrate improved performance (in terms of BLEU) over the baseline, they don't show that it is due to the effect that the authors claim to be. If we look at the uplift over the baseline in table 7, we see that this is not consitently supporting longer sequences:
> >
> > | task  | SS|E-ARM|[0-25]|uplift|[25-50]|uplift|[50-inf]|uplift|all|uplift
> > |-------| - | - |-------|------|-------|------|-------|------|------ |-
> > |De→ En | - | - | 37,72 |      | 33,24 |      | 30,86 |      | 34,61 ||
> > |       | ✔ | - | 38,20 | **0,48** | 33,76 | **0,52** | 31,08 | **0,22** | 35,10 |**0,49**|
> > |       | ✔ | ✔ | 38,37 | **0,65** | 33,92 | **0,68** | 31,43 | **0,57** | 35,36 |**0,75**|
> > |It→ En | - | - | 35,20 |      | 32,73 |      | 26,86 |      | 32,29 ||
> > |       | ✔ | - | 35,52 | **0,32** | 33,25 | **0,52** | 26,95 | **0,09** | 32,64 |**0,35**|
> > |       | ✔ | ✔ | 35,56 | **0,36** | 33,33 | **0,60** | 27,21 | **0,35** | 32,82 |**0,53**|
> > |Es→ En | - | - | 43,37 |      | 39,67 |      | 37,14 |      | 40,64 ||
> > |       | ✔ | - | 43,61 | **0,24** | 40,00 | **0,33** | 37,38 | **0,24** | 40,91 |**0,27**|
> > |       | ✔ | ✔ | 43,84 | **0,47** | 40,35 | **0,68** | 38,07 | **0,93** | 41,58 |**0,94**|
> >
> > Same conclusion can be drawn from table 8. The model with E-ARM has "just" better performance. But a slightly bigger baseline model could have had the same effect.
> > At the same time, the results with and without E-ARM are far from being state of the art (rank 17 on DE->EN leader board, cf. https://paperswithcode.com/sota/machine-translation-on-iwslt2014-german). If E-ARM is a general principle, shouldn't it be suitable to push forward state of the art?
> >
> > In summary, the method looks quite ad hoc to me. It shows performance uplift in some settings (but not exactly state of the art). It requires some empirical tuning (train only the pure ARGM for some epochs, heavily truncate backprop in $\tilde q_\theta$) which indicates that the supposed working principle (brining $p_d$ and $p_\theta$ close together) may not be 100% what's going on.
> >
> > So I would disagree that E-ARM is a "new learning pattern for autoregressive models" and consider my rating of 6 as appropriate.

---

### Official Review · Reviewer_AQxn · 2021-11-04

**Correctness:** 3
**Technical Novelty And Significance:** 4
**Empirical Novelty And Significance:** 3
**Recommendation:** 5
**Confidence:** 3

**Main Review:**

Below are the detailed strengths and weaknesses (or questions):

*Strengths*:

1. The proposed method is interesting and novel. It is also well-motivated as a new energy-based model that does not require MCMC during training.
2. The experimental results are generally good.


*Weaknesses*:

1. The proposed method only compares with vanilla autoregressive models but not other energy-based model baselines. For example, how would the method compare to (Deng et al. 2020)?  Deng et al. do not use MCMC either, thus an experimental comparison with them is important.
2. One main merit of E-ARM is its removal of the “extremely time-consuming” MCMC process, however, this merit is not supported with experimental results in this paper. I feel E-ARM can be time-consuming due to sampling from autoregressive models in Eq. 11, thus a comparison with MCMC-based baselines is preferred. In addition to efficiency, performance comparison with energy-based model baselines is also important, otherwise, I am not sure about the advantage of E-ARM compared to other energy-based methods.
3. I was confused when the authors suddenly note that the loss is summing over length K (Eq. 11). I think this point should be made earlier for clarity. For example, the sum over K should be added to Eq. 5 for correctness, right?
4. Some experimental details are missing:
(1) how many samples are used to approximate the expectations in Eq. 11?
(2) how many samples are used to approximate the weight w?
(3) do (1) and (2) share the same samples for approximation or need sampling twice?
(4) Eq. 11 is summing over K and for each k there are expectations to be approximated. This process seems very time-consuming if resampling for every k. Please clarify these details.

*Questions for the author:*

1. Is the positive phase loss in Eq. 12 also summing over K?
2. The end of page 4 says “xx means improvements in this direction will be automatically taken care of as a result of xxx”. This sentence confuses readers – I am not sure whether the first term in Eq. 7 is actually included during training or not, please clarify.
3. Appendix C says "for language modeling tasks, we fix the generation length as 50", why is generation required for perplexity computation?


**Summary Of The Paper:**

This paper designs a new globally normalized probability function to approximate the sequence data distribution, which is defined as the product of traditional autoregressive model distribution and energy-based model distribution. The authors also propose a new training objective that is different from maximizing log-likelihood. This new objective is composed of two KL divergence terms. To tackle the optimization issue due to the energy-based loss term, the authors propose to use importance sampling to sample from the autoregressive model to approximate the negative phase loss, so that MCMC is not required. Certain training tricks are required (e.g. pretraining the autoregressive model) to stabilize training.  Experiments on machine translation, language modeling, and image generation demonstrate the effectiveness of the proposed method.

**Summary Of The Review:**

The proposed method is novel and interesting, but it only compares to traditional autoregressive models in experiments, and other energy-based model baselines are missing.

---

> ### Author Response · Authors · 2021-11-22
> **Response to reviewer AQxn, about Q3-Q7**
>
> **[Q3 Loss is summing over length K]**
>
> Thanks for your advice. We are sorry for the writing of this part to cause confusion and have refined it in the latest version. And your opinion is right that the summing over length K should be added to Eq.5 to make a better understanding. In the latest version, Eq. 5 has been modified based on your kind advice.
>
> **[Q4 Missing experimental details]**
>
> 1. how many samples are used to approximate the expectations in Eq. 11/the weight w?
>
>     - The number of samples in a batch is dynamic while the maximum number of the total tokens in a batch are fixed (4096 in our experiments). If the length of sequences in a batch is 32, then it includes 4096 / 32 = 128 samples in total. It is a common strategy in language generation tasks, and has been used in many frameworks (e.g. Fairseq [https://github.com/pytorch/fairseq](https://github.com/pytorch/fairseq)).  And we generate samples autoregressively as many as the number of sequences in the current batch at each update iteration. These generated samples will be used for approximating the expectation in the negative phase of EBM training.
>
> 2. samples for approximating the expectations in Eq. 11/the weight w is shared or need sampling twice?
>
>     - We do not optimize Eq.11 directly, instead, the energy-based objective is optimized according to Eq.12. Besides, we use the same batch of samples generated autoregressively to approximate both the expectations in Eq.12 and weight w (i.e., shared), which does not need to sample twice.
>
> 3. Eq. 11 is summing over K and for each k there are expectations to be approximated. This process seems very time-consuming if resampling for every k.
>
>     - We think there are some misunderstandings about Eq. 11, for the concern of efficiency with respect to Eq. 11 (resample for every k). Although Eq. 11 contains two summation operators which may include numerous expectation calculations, it is not the final formulation to be calculated. In fact, we use Eq. 11, which is obtained by decomposing the joint distributions into a product of conditionals, to better illustrate the underlying intuition of designing the product of our EBM model and the corresponding energy function in Sec. 3. The final optimization is conducted by using the gradient of Eq. 12 for each time step k. In addition, the training of different time steps is finished in parallel, due to the benefits of Transformer. Besides, we also provide additional studies about model efficiency in Appendix D.6 in the latest version, to analyze other key factors that may affect model training efficiency, like the generation of fake sequences.
>
>
> We also add the corresponding experimental details to make a clear description. More details can refer to Appendix C.
>
> **[Q5 The positive phase loss in Eq. 12]**
>
> Yes, in each time step k, the gradients will be calculated as Eq. 12, and the back-propagation will start until the gradients of all time steps are summed up.
>
> **[Q6 The first term in Eq. 7]**
>
> The first term in Eq.7 is actually the gradient of MLE w.r.t. $\theta$, which will be optimized as the training with cross entropy loss. Since E-ARM is a union of the energy-based learning objective of Eq.12 and the cross entropy loss, the involvement of the first term of Eq.7 in training is natural and do not need to be calculated explicitly (Explicitly compute this term and involve it into the final loss is equivalent to change the hyper-parameter $\lambda$).
>
> **[Q7 Why is generation required for perplexity computation]**
>
> Sorry for the unclear description, the generation mentioned here is the fake samples sampled from the autoregressive model which is required to estimate the expectation in Eq.12. Out of the consideration of efficiency, we only generate fake sentences up to 50 tokens for language modeling which usually has input sequences with more than one hundred tokens. We have updated this paragraph in Appendix C for a clear description. The study of training efficiency can refer to Appendix D.6.
>
> [1] Yuntian Deng, Anton Bakhtin, Myle Ott, Arthur Szlam, and Marc’Aurelio Ranzato. Residual energy-based models for text generation
>
> [2] Bo Pang, Tian Han, Erik Nijkamp, Song-Chun Zhu, Ying Nian Wu. Learning Latent Space Energy-Based Prior Model
>
> [3] Max Welling, Yee Whye Teh. Bayesian Learning via Stochastic Gradient Langevin Dynamics
>
> [4] Zhisheng Xiao, Karsten Kreis, Jan Kautz, and Arash Vahdat. VAEBM: A symbiosis between variational autoencoders and energy-based models.

---

> > ### Comment · Reviewer_AQxn · 2021-11-27
> > **Response to revision**
> >
> > Thank you for clarifying many details and my questions. I feel the paper is more clear now. The added efficiency results are also appreciated. While the point that Residual ERMs use more parameters make sense, I do not think that the proposed method is different enough from other energy-based ones to excuse a comparison -- E-ARM is also proposing a new generative model because the underlying model distribution is changed, it is just E-ARM is parameterized by an autoregressive architecture but it is indeed a new generative model. In the meanwhile, other energy-based methods can be helpful to the exposure bias and incoherence problems as well.
> >
> > Therefore, I think that the proposed E-ARM is quite related to other energy-based text generation work and they belong to the same research line, and a sufficient comparison with them is necessary.

---

> > > ### Author Response · Authors · 2021-11-29
> > > **Response to reviewer AQxn**
> > >
> > > Thanks for your comments, we will report the results of comparison once current experiments finished.

---

> ### Author Response · Authors · 2021-11-22
> **Response to reviewer AQxn, about Q1-Q2**
>
> Thanks for your valuable comments. Below are our responses to your comments:
>
> **[Q1 Comparisons with other energy-based models]**
>
> Thanks you for suggestions. We have noticed some related works, and the below are some results of comparisons:
>
> - For energy-based methods, we currently conduct experiments to compare our E-ARM with Residual EBMs [1]. Considering the limitation of computing resources and time, the experiments of our E-ARM performance on the two benchmarks (CC-News, Toronto books) reported by [1] is still undergoing, since [1] has applied 8 DGX nodes and each with 8 Nvidia V100s. We will report the results on these two datasets when it is finished in the future. To make a fast verification,  we have implemented their Residual EBMs on the Wikitext103 dataset based on transformer-xl network and achieved 23.96 PPL, which is slightly worse than the performance 23.81 PPL of our E-ARM with transformer-xl. This results can show the effectiveness of our E-ARM is comparable with Residual EBMs. Besides, the Residual EBMs requires an extra energy estimation network which needs the same learnable parameters as the base autoregressive networks, while our method only uses single autoregressive network (i.e., their model requires 2 times parameters as ours). We will updated the results once the rest experiments are finished.
>
> As we aforementioned, our method is more likely to a novel training pattern for training a better autoregressive model to solve the exposure bias and incoherence problems, without introducing any further learnable parameters or special architectures. As a result, the comparison with methods which solve exposure bias, like scheduled sampling listed in our paper, is more reasonable compared to those methods, which proposed a new generative model (VAEBM [4], Latent Space Energy-Based Prior Model[1], etc).
>
> **[Q2 Comparison with MCMC-based baselines]**
>
> Thank you for pointing out this issue. To address your concerns, we have provided the analysis and comparisons between MCMC-based methods in terms of efficiency in Appendix D.6.  Generally, sampling fake data for the negative phase of energy-based training is a key part of the optimization of an EBM. In our work, we design a unique way to avoid the MCMC sampling from the joint distribution $p_\theta(x_k, \textbf{x}_{<k})$ in the energy-based optimization process by generating samples from the ARGM. There are two main reasons for this: (1) we argue that generating samples autoregressively is a more efficient process compared to MCMC sampling. (2) for sequential data like text, the intrinsic discrete property prevents it from applying MCMC in the data space directly.
>
> In Appendix D.6, due to the second reason mentioned above, we applied SGLD [3] sampling (which is one of MCMC methods) at the first latent layer, and Eq.5 is optimized with Eq.6 directly (Eq.6 requires sampling data from the product EBM $p_\theta(x_k, \textbf{x}_{<k})$ , which needs MCMC). According to our analysis and experiments, the training time cost of our E-ARM  is approximately 5 times as large as the vanilla training, and the training efficiency of E-ARM is limited by the auto-regressive decoding. In contrast, the training time cost of optimizing with SGLD sampling is nearly 100 times as large as the vanilla training, and even using short-run SGLD (k = 20 with a sacrifice of performance), it still requires roughly 20 times training cost than the standard training. As a result, optimizing the energy-based learning objective by generating samples autoregressively is a more efficient way than sampling data by MCMC sampling. (More detailed analysis and experiments are introduced in Appendix D.6)

---

### Official Review · Reviewer_in11 · 2021-11-08

**Correctness:** 3
**Technical Novelty And Significance:** 3
**Empirical Novelty And Significance:** 3
**Recommendation:** 6
**Confidence:** 5

**Main Review:**

The paper is well-written and easy to follow. The results look promising and the idea of using EBM guidance in AR model training is interesting. I have some concerns as follows:

* Although the importance sampling avoids MCMC from the EBM, it does require sampling from $\tilde{q}(x_{<k})$ which goes through the autoregressive generation process and could be time-consuming. Would be informative if you could report FLOP in the paper. However, I do feel that it is way more complicated than a simple regularizer added on to the training of the autoregressive model.

* I still doubt that high variance is induced in the importance sampling as it is not clear how close $p_\theta$ and $\tilde{q}_\theta$ and another Monte Carlo estimate is required when computing the importance weight $w$. Do you observe training instability during training? Would be better to provide more analysis on this point.

* It would be more informative if you could report the results before the Top-K correcting by EBM in the inference stage, as it would clearly show if the AR model really benefits from the joint training.

* In the paper, it is claimed that the joint training improves the long-range coherence. Is there a metric to show this point empirically?

* From table 4, it looks like the optimal solution requires a very small weight of the EBM KL. For what reason that a larger weight harms the performance? Theoretically, it should work with even only the EBM KL.

Minor:
- Typo for "The first term *** in Eq. 7 is equivalent to ..." in the line after Eq.8: should be $-\mathbb{E}_{x<k \sim p_d}$
- Here is a related work of using NCE to learn EBM that you should consider to discuss or cite in the related work:
"Flow Contrastive Estimation of Energy-Based Models", CVPR 20.


**Summary Of The Paper:**

This paper proposes to improve the training of AR generative models by introducing a "regularizer" based on a decently designed energy-based model. The EBM is seamlessly integrated with the AR model and the sampling from the EBM can be transformed as an importance sampling from the AR model. Empirical results on text and image datasets verify the effectiveness of the proposed model.

**Summary Of The Review:**

Overall speaking the paper is well-written and well-motivated. The direction of joint training of AR model and EBM is promising. However, I'm sort of concerned with the computational cost and training instability introduced by the extra EBM module, compared to the performance gain it brings.

---

> ### Author Response · Authors · 2021-11-22
> **Response to reviewer in11, about Q4-Q7**
>
> Thanks for your valuable comments. Below are our responses to your comments:
>
> **[Q4 Metrics for long-range coherence]**
>
> Thank you for your suggestions. We have conducted a series of experiments to measure the ability of our E-ARM in solving long-range coherence. Specifically,  we divide the test set of IWSLT14 (German $\rightarrow$ English, Italian $\rightarrow$ English, Spanish $\rightarrow$ English) translation dataset into three subsets ([0, 25], [25, 50], and [50, $\infty$)) based on the target sentence lengths. The results are shown in Appendix D.1. Then, we respectively evaluate scheduled sampling and our E-ARM based on transformer base network, and test their performance on these three different subsets. Generally, the subset of samples with longer sentences will be more affected by the long-range incoherence problem. We observe that, after applying our E-ARM together with the scheduled sampling technique, the base model can further obtain additional performance gain. Specifically, the improvement on the longer sentence is more evident, since the model can obtain large improvements on the [50, $\infty$) (e.g. On German to English task, 31.43 - 31.08 = +0.35 points for [50, $\infty$) test sets) than short sets [0, 25] and [25, 50] (e.g. On German to English task, 38.37 - 38.20 = +0.17 points and 33.92 - 33.76 = + 0.16 points for [0, 25) and [25, 50) test sets respectively). This phenomenon indicates that our E-ARM can resolve the incoherence problem to some extent. Please refer to Appendix D.1 in our latest version for more details and discussion.
>
> **[Q5 Reason why a larger weight harms the performance]**
>
> Thank you for your question. We think this question is related to Q2. As aforementioned in the response of Q2, the single optimization of the energy-based learning objective is problematic, since the support of real sequence $p_d$ is disjoint to the support of distribution $p_\theta$ at the beginning of the training process. Although optimizing the EBM learning objective alone is feasible in such a scenario where we obtain a well-trained autoregressive model first and then apply the EBM learning objective to further finetune the model. However, this two-stage training strategy does not obtain better performance when compared to our E-ARM in Table 3 (such two-stage training obtains 24.11 PPL for language modeling task on Wikitext103 dataset with Standard Tr-XL model, while E-ARM can achieve 23.81 under the same setting). We opine this is because once the network has achieved convergence in the vanilla training manner, it is hard for the model to find a way to climb out the local minimum and thus can be benefited less by our EBM training. Moreover, when it comes to the coefficient $\lambda$, the value of it depends on the gradients' magnitudes of two parts and we need to balance the ratio between two parts' magnitudes. Therefore, we use a relatively small $\lambda$ because the gradient of the EBM objective is large, and the training under such a $\lambda$ is stable (See appendix D.3).
>
> **[Q6 Typo for Eq. 7]**
>
> Thanks for the correction, we have fixed those typos in the latest version.
>
> **[Q7 Related works of using NCE to learn EBM]**
>
> Thanks for your suggestions, we have cited this paper and included the corresponding discussion of related work in the new version.

---

> > ### Comment · Reviewer_in11 · 2021-11-22
> > **Thank you for the reply**
> >
> > Thanks the response from the author(s).
> >
> > I appreciate the additional experiments the author(s) conducted in Appendix D and the correction in Section 4, which address my concerns well. I will change my score to 6.

---

> ### Author Response · Authors · 2021-11-22
> **Response to reviewer in11, about Q1-Q3**
>
> Thanks for your valuable comments. Below are our responses to your comments:
>
> **[Q1 Time-consuming for sampling from q(x_<k)]**
>
> Thank you for pointing out this issue. We have added related analysis and experiments in Appendix D.6.  In general, sampling fake data for the negative phase of energy-based training is a key part of the optimization of an EBM. In our work, we design a unique way to avoid the MCMC sampling from the joint distribution $p_\theta(x_k, \textbf{x}_{<k})$ in the energy-based optimization process by generating samples form the ARGM. There are two main reasons for this: (1) we argue that generating samples autoregressively is a more efficient process compared to MCMC sampling; (2) for sequential data like text, the intrinsic discrete property prevents it from applying MCMC in the data space directly.
>
> In Appendix D.6, due to the second reason mentioned above, we applied SGLD [1] sampling (which is one of MCMC methods) at the first latent layer, and Eq.5 is optimized with Eq.6 directly (Eq.6 requires sampling data from the product EBM $p_\theta(x_k, \textbf{x}_{<k})$, which needs MCMC). According to our analysis and experiments, the training time cost of our E-ARM  is approximately 5 times as large as the vanilla training, and the training efficiency of E-ARM is limited by the auto-regressive decoding. In contrast, the training time cost of optimizing with SGLD sampling is nearly 100 times as large as the vanilla training, and even using short-run SGLD (k = 20 with a sacrifice of performance), it still requires roughly 20 times training cost than the standard training. As a result, optimizing the energy-based learning objective by generating samples autoregressively is a more efficient way than sampling data by MCMC sampling. (More detailed analysis and experiments are introduced in Appendix D.6)
>
> Besides, we agree that viewing E-ARM as a regularizer is an inappropriate description, and thus we have modified this part in section 4 as "In general, E-ARM ought to be viewed as a new learning pattern for autoregressive models that ensures our base autoregressive network stays close to the real distribution" in the latest version.
>
> **[Q2 High variance in the importance sampling]**
>
> Thank you for your valuable comments and for pointing out the problems in the variance.  We admit that our analysis about variance has some mistakes in the original version, and we have fixed it now. The actual reason for us to not train the energy-based learning objective alone is that at the initial stage of the training process, what we have is a randomly initialized autoregressive model (e.g., Transformer in our experiments), which outputs meaningless sequences when given any context. This results in disjoint supports between the real sequence's distribution $p_d$ and distribution $p_\theta$ modeled by autoregressive networks at the beginning. If we optimize the second part of Eq.5 solely, the gradient
> $\mathbb{E}_{p_d} [\frac{\partial}{\partial \theta} \log p_\theta(x_k, \textbf{x}_\{<k\}) ]$
>
> would be 0 due to the disjoint supports between $p_d$ and $p_\theta$, which means long-term unstable training and an extremely small convergence rate. Besides, the variance in the importance sampling is not a problem of our method and will not affect our final results, since the importance weight in our method would be 1 for all $x_{<k}$ at the beginning of the training. Thanks again for pointing out this mistake and we have fixed this part in the latest version (See the penultimate paragraph in Section 4).
>
> **[Q3 Results before Top-K correcting by EBM]**
>
> Thanks for your suggestions. We have added the corresponding experiments in Appendix D.4. We have studied the effect of different K to model performance. When K is 0, it means the model does not use Top-K sampling (Top-K correction). We find the performance of model can be improved by Top-K sampling, however, the performance boost is minor. These results also indicate the performance improvements of E-ARm are mainly from the joint-training, rather than Top-K energy re-sampling.
>
> [1] Max Welling, Yee Whye Teh. Bayesian Learning via Stochastic Gradient Langevin Dynamics

---

### Official Review · Reviewer_FnWE · 2021-11-08

**Correctness:** 3
**Technical Novelty And Significance:** 2
**Empirical Novelty And Significance:** 1
**Recommendation:** 3
**Confidence:** 4

**Main Review:**

Strength:
+ This paper is well-written and well-organized. Issues on conventional ARGMs are clearly pointed out, which lead to the motivation of this work.
+ Three experiments show the overall improvement of E-ARM compared with based ARGM.

Weakness:
- Training ARGM with EBM in a cooperative manner has early been proposed by
Learning Latent Space Energy-Based Prior Model, NeurIPS 2020
Cooperative Training of Descriptor and Generator Networks, TPAMI 2018
Cooperative Training of Fast Thinking Initializer and Slow Thinking Solver for Conditional Learning, TPAMI 2021
These works, however, are essentially missed from related works and model comparisons.
- The experiments of this work are relatively insufficient.
  1. All experiments only demonstrate improvements compared with base model, while comparisons with other baseline models are missing.
  2. Only basic evaluation metrics are used. Some other metrics, e.g. ROGUE, METEOR for NMT, FID for image generation, should also be listed.


**Summary Of The Paper:**

This work proposes E-ARM, aiming at integrating EBM into ARGM seamlessly in order to tackle two major existing issues on sequence modeling, i.e., exposure bias and incoherence problems. At each step, the joint distribution $p(x_k, x_{<k}$ is optimized by not only autoregressive cross-entropy loss, but also the EBM KL divergence. Moreover, the model uses importance sampling to bypass the MCMC as conventional EBM models do. Experiments on neural machine translation, language modeling and image samling all demonstrate that the proposed integration outperforms the base model.

**Summary Of The Review:**

This work proposes to alleviate ARGM existing problems by leveraging the EBM objective. Such integration, however, is not essentially new and this work does not demonstrate its superiority compared with other methods.  Despite three experiments performed in this work, the evaluation and analysis are insufficient.

---

> ### Author Response · Authors · 2021-11-22
> **Response to reviewer FnWE, about Q2-Q3**
>
> Thanks for your valuable comments. Below are our responses to your comments:
>
> **[Q2 Comparisons with other baseline models]**
>
> Thanks you for suggestions. We have noticed some related works, and the below are some results of comparisons:
>
> - We have conducted an experiment on the Penn Treebank dataset (PTB) for language modeling tasks to compare our model with [1] although these two works are different. As a result, we obtain 53.83 PPL based on the transformer-xl network, for reference. Besides, we also notice a similar work, named as Residual EBMs [8], which also attempt to solve the exposure bias on language modeling. Therefore, we currently conduct experiments to compare our E-ARM with Residual EBMs [8]. Considering the limitation of computing resources and time, the experiments of our E-ARM performance on the two benchmarks (CC-News, Toronto books) reported by [8] is still undergoing, since [8] has applied 8 DGX nodes and each with 8 Nvidia V100s. We will report the results on these two datasets when it is finished in the future. To make a fast verification,  we have implemented their Residual EBMs on the Wikitext103 dataset based on transformer-xl network and achieved 23.96 PPL, which is slightly worse than the performance 23.81 PPL of our E-ARM with transformer-xl. This results can also show the effectiveness of our E-ARM is comparable with Residual EBMs. Besides, the Residual EBMs requires an extra energy estimation network which needs the same learnable parameters as the base autoregressive networks, while our method only uses single autoregressive network (i.e., their model requires 2 times parameters as ours). We will updated the results once the rest experiments are finished.
>
> As we aforementioned, our method is more likely to a novel training pattern for training a better autoregressive model to solve the exposure bias and incoherence problems, without introducing any further learnable parameters or special architectures. As a result, the comparison with methods which solve exposure bias, like scheduled sampling listed in our paper, is more reasonable compared to those methods, which proposed new generative models (VAEBM [9], Latent Space Energy-Based Prior Model[8], etc).
>
> **[Q3 Other evaluation metrics]**
>
> Thank you for your ideas. In Appendix D.5, we have expanded the evaluation metrics for translation tasks to include ROUGE-1, ROUGE-2, ROUGE-L, METEOR plus BLEU. We adopt BLEU to evaluate translation jobs in the original paper because it is a widely-used metric to measure the performance of translation tasks in many classic publications [4,5,6], and so the BLEU metric is a convincing metric to represent model quality. And metrics like ROUGE are commonly employed to solve summarization problems [7]. Since these metrics can further prove the effectiveness of our method, we also conduct experiments to evaluate these metrics and the results in Appendix D.5 also demonstrate the same conclusion that our method can improve model performance and alleviate the exposure bias/long-range coherence problems in the autoregressive generation.
>
> [1] Bo Pang, Tian Han, Erik Nijkamp, Song-Chun Zhu, Ying Nian Wu. Learning Latent Space Energy-Based Prior Model
>
> [2] Jianwen Xie, Yang Lu, Ruiqi Gao, Song-Chun Zhu, Ying Nian Wu. Cooperative Training of Descriptor and Generator Networks
>
> [3] Jianwen Xie, Zilong Zheng, Xiaolin Fang, Song-Chun Zhu, Ying Nian Wu. Cooperative Training of Fast Thinking Initializer and Slow Thinking Solver for Conditional Learning
>
> [4] Ashish Vaswani, Noam Shazeer, Niki Parmar, Jakob Uszkoreit, Llion Jones, Aidan N. Gomez, Lukasz Kaiser, Illia Polosukhin. Attention Is All You Need
>
> [5] Ilya Sutskever, Oriol Vinyals, Quoc V. Le. Sequence to Sequence Learning with Neural Networks
>
> [6] Dzmitry Bahdanau, Kyunghyun Cho, Yoshua Bengio. Neural machine translation by jointly learning to align and translate
>
> [7] Chin-Yew Lin. ROUGE: A Package for Automatic Evaluation of Summaries
>
> [8] Yuntian Deng, Anton Bakhtin, Myle Ott, Arthur Szlam, and Marc’Aurelio Ranzato. Residual energy-based models for text generation
>
> [9] Zhisheng Xiao, Karsten Kreis, Jan Kautz, and Arash Vahdat. VAEBM: A symbiosis between variational autoencoders and energy-based models.

---

> > ### Comment · Reviewer_FnWE · 2021-12-07
> > **Concerns still exist for experiments.**
> >
> > Thank the authors for their careful rebuttal. However, after reading your response and revision, I still feel the current experiments are too weak to justify the superiority of E-ARM. For instance, Table 1 only compares the proposed model with base ARGM, while the rest EBM-based models are not included.

---

> > > ### Author Response · Authors · 2021-12-07
> > > **Response to Reviewer FnWE**
> > >
> > > We will report the results of comparison between our method and residual EBM[8] once current experiments finished, and update it in the final version of our paper. The experiment in [8] requires rather heavy overheads (they have applied 8 DGX nodes and each with 8 Nvidia V100s) and is very time-consuming.
> > >
> > > [8] Yuntian Deng, Anton Bakhtin, Myle Ott, Arthur Szlam, and Marc’Aurelio Ranzato. Residual energy-based models for text generation

---

> ### Author Response · Authors · 2021-11-22
> **Response to reviewer FnWE, about Q1**
>
> **[Q1 Missed related works]**
>
> Thank you for your notifications. We are sorry for the missing citations of these works and have included them in the latest version of our paper, with the corresponding discussions (Please see the appendix. E). For these works, we want to highlight that our work has the following differences:
>
> - [1] aimed to learn an energy-based model in the latent space of a generator model, so that the EBM can act as a prior model on the generator model’s top-down network. The target of [1] is that EBM in the latent space can capture regularities and thus benefit the training of the generator model. In contrast, by carefully constructing an energy-based learning objective and its corresponding optimization procedure, E-ARM is able to smoothly integrate energy surface learning into autoregressive networks without any additional learnable parameters. We want to highlight that the target of our E-ARM is to improve the performance of autoregressive models and address its intrinsic characteristics (e.g., exposure bias or lack of long-range coherence) by using a well-designed energy-based learning objective, instead of proposing a novel  generative models. Therefore, our method and [1] have differences in essential, including method, motivation and tasks. Besides, we have evaluated our E-ARM on the Penn Treebank dataset (PTB) for language modeling tasks as a fast validation. As a result, we obtain 53.83 PPL based on the transformer-xl network, for reference. Due to the time limitation, we will add more comparisons in the final version.
> - [2, 3] are mainly to learn the conditional distribution of a high-dimensional output given input by combining the efforts of a fast-thinking initializer, which generates the output, a latent vector, and a slow thinking solver, which learns an objective function in the form of a conditional energy function. The outputs can be generated by optimizing the objective function, or more rigorously by sampling from the conditional energy-based model. Both of them requires an additional network to learn the energy scores while ours do not need. And both of them do not allow the autoregressive model itself to enjoy the benefits from EBM in modeling the joint distribution while it's the main goal of our method. Therefore, these two works are different from our E-ARM. We have added corresponding discussions in Appendix. E in the latest manuscript.
>
> Our method is more likely to a novel training pattern for training a better autoregressive model to solve the exposure bias and incoherence problems, without introducing any further learnable parameters or special architectures. We attempt to benefit the base autoregressive network itself with the introduced energy-based learning objective. More discussions about more related works have been appended in the latest version (Please see Appendix. E).

---

### Public Comment · ~Marc_Dymetman1 · 2021-11-12
**Interesting paper, calling attention to related work in this area**

Dear authors,

I read your paper with much interest and would like to call to your attention a line of work [1,2,3] that you do not currently cite but which, while being clearly different from what you propose, is also, uncontroversially IMO, relevant related work.

[1,2] were among the first papers, along with and cited by (Deng et al. 2020, Bakhtin et al. 2021), to combine EBMs with ARGMs in order to reduce the perplexity of the combined model over the training set, and with the exact same purpose as your title. Experiments were done on simulated data, and [3] later extended these models to impose prescriptive (as opposed to descriptive, as in [1,2]) distributional controls on realistic text generation tasks, in particular for social debiasing purposes.

It is not here the place to go into a detailed comparison between [1,2,3] and other EBM models for sequential data. However, in regard to the last paragraph in your related work section [“As a result … strategy of training an EBM without MCMC …”] I would like to stress that [1,2,3] also, and centrally, do not use MCMC, but give a core role to importance sampling in particular at a point where they perform “moment matching” between the data and the EBM as well as inside their Distributional Policy Gradient (DPG) algorithm for fine-tuning the ARGM towards the underlying EBM.

Cheers,

Marc Dymetman
NAVER Labs Europe

[1] T Parshakova, JM Andreoli, M Dymetman:
Global Autoregressive Models for Data-Efficient Sequence Learning. CoNLL 2019 https://aclanthology.org/K19-1084/

[2] T Parshakova, JM Andreoli, M Dymetman:
Distributional Reinforcement Learning for Energy-Based Sequential Models. https://arxiv.org/abs/1912.08517 (2019)

[3] M Khalifa, H Elsahar, M Dymetman:
A Distributional Approach to Controlled Text Generation. ICLR 2021 https://openreview.net/forum?id=jWkw45-9AbL

---

> ### Author Response · Authors · 2021-11-22
> **Response to Marc Dymetman**
>
> Thanks for your notification, we have cited these related works, and added the corresponding discussions about these works in the latest manuscript.

---

### Decision · Program_Chairs · 2022-01-20

**Decision:**

Reject

**Comment:**

This paper proposes a method to train autoregressive model that takes advantage of a well-designed energy-based learning objective model. With the importance sampling, the model can be trained efficiently without requiring an MCMC sampling. Experiments are conducted to verify the effectiveness of the proposed method.  The idea is interesting and well-motivated, but the experiments need to be improved. Reviewer FnWE’s major concerns include limited novelty, lack of discussion with closely related works, and insufficient experiments, and recommend rejecting the paper by assigning a rating of 3. Rebuttal doesn’t address his/her concerns. Reviewer in11 is concerned with the computational cost and training instability due to the extra EBM module and has a few unclear technical details that need to be clarified. The author’s reply along with additional experiments during rebuttal partially addresses the concerns of Reviewer in11, who eventually increases the rating to 6. Reviewer AQxn’s major concern is also about the lack of sufficient comparison with other relevant energy-based models.  Reviewer DZsJ pointed out that the more insightful analysis about the model is missing in the experiments. Even though the authors provide additional experiments for Reviewer DZsJ, they are not satisfied with the feedback because the additional results are not supportive of the claims made in the paper, and end up with a rating of 6. Reviewer SjXn’s concerns include the lack of comparison with relevant works and the unclear motivation of the design of the joint distribution. After the rebuttal, Reviewer SjXn’s concerns remain and assign a rating of 5 to the paper. The overall rating of the paper after rebuttal is marginally below the acceptance rate. Even though this paper proposes an interesting idea, the reviewers’ comments are not well addressed. As a result, AC cannot recommend accepting the paper.  The AC urges the authors to revise their paper according to the comments from the reviewers, and resubmit their work in a future venue.